# Getting robots back on track by reconstituting control in unexpected situations with online learning

Maxime Allard [1,2] ✉, Manon Flageat [1,2], Bryan Lim[1] & Antoine Cully [1]

Robotic systems are increasingly common in strictly controlled environments (e.g. warehouses), but they have yet to fully integrate into our everyday lives. Everyday scenarios are a challenge for robot controllers due to disturbances that can be caused by unforeseen conditions (e.g., damage, flat tires, or wind gusts) or unexpected usage. In these cases, operators may lose control of their systems, leading to potentially catastrophic outcomes, such as crashes or failed operations. Our method, "Fast Learning-based Adaptation for Immediate Recovery", uses machine learning to counter loss of control by enhancing existing controllers. It rapidly diagnoses and compensates for unseen perturbations by updating an onboard model every 225 milliseconds. Even in a setting with only onboard compute, we show that operators equipped with our method regain control and operate as effectively as in unperturbed conditions across a wide range of perturbations. Other state-of-the-art methods, such as an optimal control and an adaptive control baseline, were found to be half as effective at recovering from perturbations, while an online Deep Reinforcement Learning baseline proved entirely ineffective. In this work, we demonstrate that our online learning method enhances robotic resilience by mitigating the impact of perturbations on system operability.

Robots and vehicles are meant to operate in various challenging situations[1–4], where they might face unexpected perturbations such as unusual friction conditions, reduced motor running torques, changes in weight distributions, flat tires, wind gusts and many more. A robotic system is operable when it behaves according to the operator's expectations for a given command, as depicted in Fig. 1C, D. However, given desired commands, unexpected perturbations can cause vehicles to exhibit behaviours that deviate from the originally designed behaviours. A system is inoperable if the behaviour induced by a command does not follow those expectations.

The loss of operability can have potentially catastrophic consequences. An example of this is the unfortunate event of March 2021, where a container ship (Ever Given) blocked the Suez Canal for six days. This was caused by perturbations in the form of high-winds and a dust storm, which consequently resulted in strong Bernoulli forces acting upon the ship while navigating through the canal[5], leading to worldwide delays (see Fig. 1A). Ensuring and maintaining the operability of robots and intelligent vehicles under perturbations is an important step towards their integration into our everyday lives and operations.

Many prior methods to this problem have been concerned with designing controllers that aim to maintain operability under a pre-specified set of perturbations. This can be achieved by performing trajectory optimisation[6], fast re-planning through model-predictive control approaches[7–9], or explicit modelling of perturbation scenarios[10,11]. However, such approaches quickly increase in complexity when more sources of perturbation are considered, and their performance can degrade if the robot encounters scenarios not pre-specified[12].

Similarly, Reinforcement Learning (RL)[13] has aimed to simplify the manual design process of controllers, and can result in controllers that demonstrate behaviours that cannot be modelled, designed or

[1]Adaptive And Intelligent Robotics Lab, Department of Computing, Imperial College London, Exhibition Rd, London SW7 2BX, United Kingdom. [2]These authors contributed equally: Maxime Allard, Manon Flageat. ✉e-mail: m.allard20@alumni.imperial.ac.uk

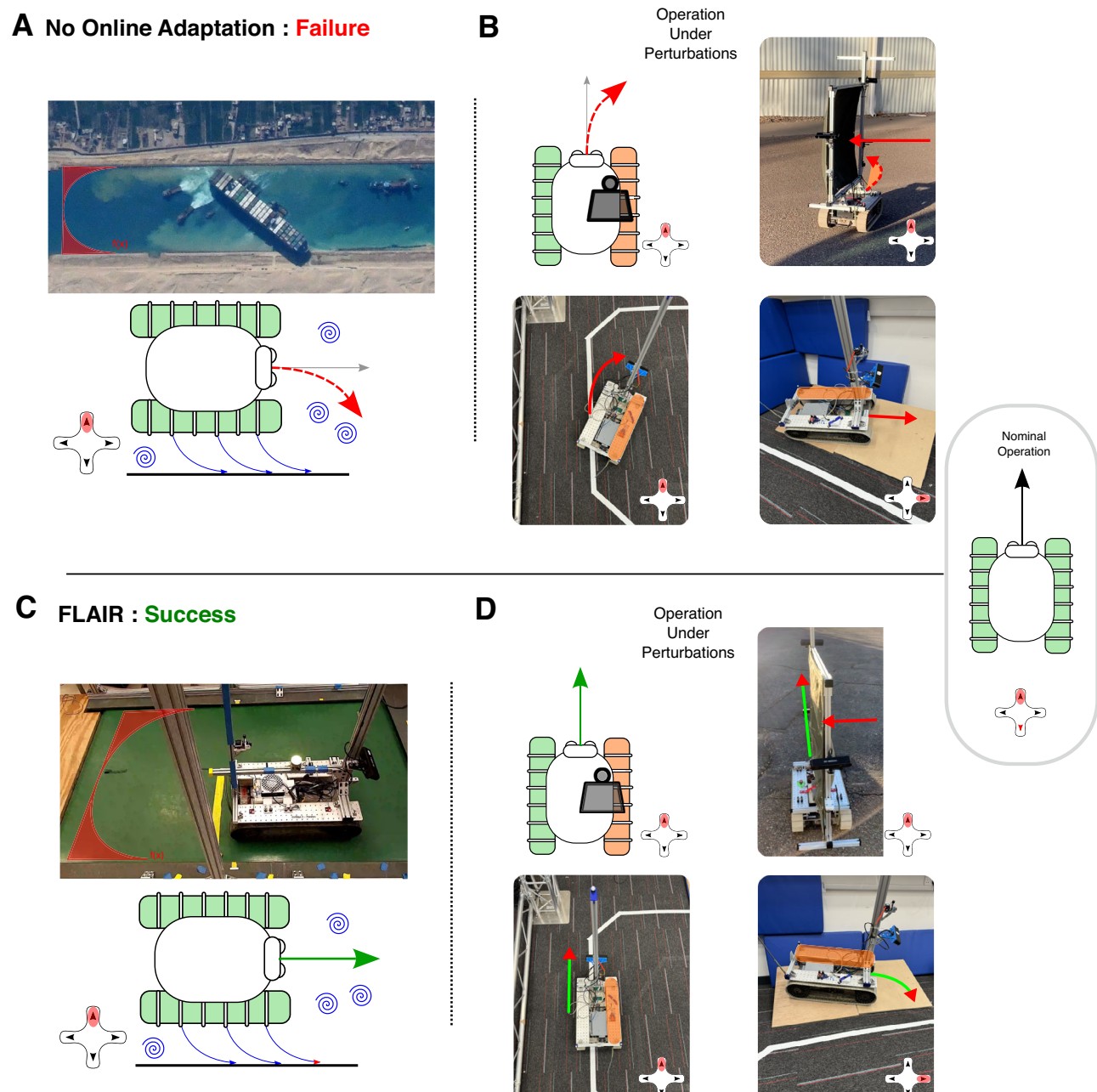

**Fig. 1 | Challenges of operating vehicles under perturbations and the benefits of FLAIR.** Each panel shows a control pad with the operator command highlighted in red, and the robot's behaviour as an arrow. In our experiments, we use a tracked mobile robot to test our method in different situations. **A, B** When perturbations occur, vehicles might become inoperable and not behave as commanded (grey arrow), resulting in unexpected outcomes (red arrow). **C, D** FLAIR enables vehicles to overcome those perturbations and correctly follow commands as in nominal operation (green arrow), allowing the operator to remain in control when the system is perturbed. **A** A notorious example of perturbations is Bernoulli forces that can pull a ship toward the shore of a canal. An example of such an event is the Ever Given ship that was stranded in the Suez Canal. Image courtesy of the Earth Science and Remote Sensing Unit, NASA Johnson Space Center (https://eol.jsc.nasa.gov/SearchPhotos/photo.pl?mission=ISS064roll=Eframe=48480). **C** We re-created the canal conditions on a treadmill by applying high perturbations near the border. **B** Similarly, the operation of ground vehicles can be heavily impacted by perturbations such as weight distributions or wind gusts. **D** FLAIR is able to rapidly learn the effects of all perturbations and counteract them, reconstituting control for the operator.

engineered[14–16]. However, being a data-driven approach, RL requires controllers to be optimised beforehand using a large amount of interaction with the environment under different perturbations to ensure the controller remains operable when facing them.

Both these families of approaches rely on designing new controllers to handle any additional type of perturbation. Most similar to our approaches are online learning methods which adapt while collecting data using trials during deployment[17–22]. However, these methods require minutes[20,23,24] to hours or days[17] to adapt.

As another line of work, fault-tolerant control focuses on detecting faulty sensors and recovering from their impact once isolated[25]. To this end, mathematical models are learned to generate residual signals that can be used to detect a fault[26], and combined with Active Inference[27–30] to reach a desired state[27,31]. However, because they rely on explicit fault estimation, these methods typically address specific types of perturbations with respect to sensors.

Alternatively, the literature around adaptive control and stochastic control tries to account for uncertainty by adjusting the

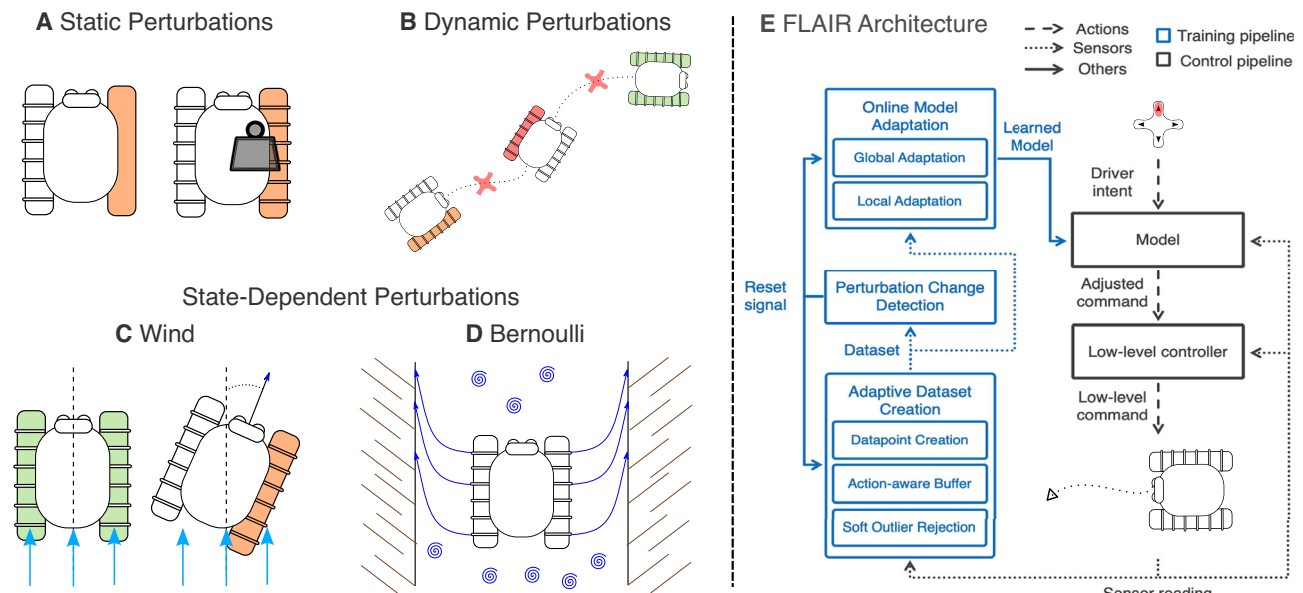

**Fig. 2 | Evaluation perturbations. A** Static perturbations, such as track degradation or change in weight distribution, impact the dynamics of the system. **B** These perturbations can also be dynamic and change over time, for example, if the robot gets out of a difficult situation. Additionally, we tested complex state-dependent perturbations such as (**C**) the wind pushing the robot to the side, where the blue arrows represent the wind direction, or (**D**) the Bernoulli effect in a canal, where the dark blue arrows represent the impact of the Bernoulli force. **E** Flowchart of the FLAIR architecture.

internal controller characteristics. Methods such as $\mathcal{L}_1$ Adaptive Control[32] use reference models to predict the next best actions. However, these methods adopt a reactive approach with a reference model, which might fail to adapt to complex perturbations if the model is mismatched or the disturbances are highly nonlinear. More details in Supplementary Related Work.

In this work, we propose a hierarchical learning method that augments the existing controller and adjusts the commands it receives to ensure the robot remains operable. Our method, "Fast Learning-Based Adaptation for Immediate Recovery" (FLAIR), learns this adaptation online from raw data, using only onboard computation and sensors, while the robot is controlled by an operator. Additionally, FLAIR offers the operator detailed insights into the perturbation affecting the robot, allowing crucial introspection abilities for quick situation diagnosis.

This paper demonstrates that the hierarchical addition of an abstract layer with spatio-temporal data processing on top of the robot's original black-box controller enables rapid adaptation of differential-drive robots to a variety of perturbations. The experiments consider both deployed tracked vehicles and simulated hexapod robots, facing perturbations including low-friction grounds, scaled motor torques, and external perturbations such as an artificial wind (illustrated in Fig. 2). Our results show not only that a learning-based controller can adapt to external perturbation in an online manner using exclusively onboard compute, recovering 74.4% of the operability of the unperturbed robot, but also outperforms both optimal control and adaptive control baselines in our experiments by a factor of 2. Furthermore, by taking a hierarchical approach to adaptive control, we enable introspection capabilities, using a simple (e.g., polynomial) top-level model to represent the effect of perturbations.

In addition to the experiments in this manuscript, FLAIR has also been deployed and tested during the Learning Introspective Control (LINC) program organised by DARPA to respond in real-time to perturbations not taken into consideration at design time.

### Fast learning-based adaptation for immediate recovery
**Problem statement.** Perturbations acting upon a robot alter its behaviour $\mathbf{b} \in \mathcal{B}$ (in our case a vector with the velocities $v_x$ and $\omega_z$ for a

given command vector $\mathbf{a} \in \mathcal{A}$. When faced with such perturbations, operators attempt to learn a mental model on how to change actions $\mathbf{a}$ into $\mathbf{a}' \in \mathcal{A}$ to obtain the desired behaviour $\mathbf{b}$ and regain operability (i.e., obtaining desired behaviour $\mathbf{b}$ through a command). In this work, we formalize a perturbation as a function $f(\mathbf{a}, \mathbf{s}) = \mathbf{b}'$ that modifies the robot's behaviour when actions $\mathbf{a} \in \mathcal{A}$ are executed in state $\mathbf{s} \in \mathcal{S}$ under the perturbation. Our goal is to learn $f$ in order to counteract the perturbation and augment the operator:

$$f : \mathcal{A} \times \mathcal{S} \to \mathcal{B}' \qquad (1)$$

**Hierarchical mapping.** To reconstitute control, FLAIR learns a hierarchical command-behaviour mapping that characterises the effect of perturbations on the system kinematics and the behaviour of the robot. The overall FLAIR pipeline is illustrated in Figure 2E.

When learning the model, FLAIR decomposes perturbations into global ones, affecting the entire range of commands similarly (e.g., constant change in motor torque range) and local ones, affecting only certain independent commands (e.g., slippage). Accounting for local perturbations enables a more fine-grained modelling of complex perturbations. This model is based on GPs, trained online on the onboard computer. Importantly, FLAIR learns entirely from online data collected through the robot's sensors and trains a new model for every new unseen situation in less than 225ms. To achieve this, it maintains an adaptive dataset using a set of dedicated modules that limit data redundancy, constrain dataset size, ensure sufficient coverage of the command space, and reduce uncertainty. Finally, to handle highly dynamic scenarios, FLAIR autonomously detects changes in perturbations, triggering model and dataset resets, thus enabling the learning of a new model within seconds.

## Results
We conduct real-world experiments using a tracked mobile robot displayed in Fig. 1 and shown in Supplementary Movie 1. The robot comes with a controller that accepts linear velocity and angular velocity commands and outputs motor velocities to move the tracks. We also illustrate the generalisability of our method using a

simulated hexapod robot whose low-level controller is trained following Cully et al.[20].

## Challenges of unexpected situations

FLAIR has been designed to augment the operator's capabilities by automatically and rapidly rejecting complex unseen (i.e., not trained on) perturbations to provide the feeling of nominal and regular operation. Without this rejection, the operator would have to manually compensate for the perturbations (e.g., over steering, increasing acceleration), which increases the mental load of the operator.

In the following, we define three general types of perturbations, inspired by real-world scenarios, that are used in our experiments (illustrated in Fig. 2). First, a static perturbation scenario (Fig. 2A) which simulates perturbations that remain constant throughout deployment, such as a shifted centre of mass (CoM) or a degraded track[33,34]. Second, a dynamic perturbation which changes over the course of the deployment (i.e., time) (Fig. 2B), for example, occurring when a robot moves through various terrains with different friction coefficients. These perturbations test the capability to adapt to new situations that have not been learned by the model, mimicking the real-world.

Finally, the third type of perturbation changes over time but is also state-dependent. Examples include continuous wind blowing into the robot (rotation-dependent) or closeness to borders for Bernoulli forces (position-dependent).

On top of these three controlled evaluation settings, there are also perturbations that occur naturally, such as high slippage of tyres and tracks or vibrations. For example, the robot we use in this paper does not have a uniform weight distribution as the batteries are located on the left side, shifting the CoM towards the left track.

In our experiments, the three types of perturbations are simulated by scaling the velocity of the left or right track of the robot with scaling factors $d_{left}, d_{right} \in [0, 1]$, information that is not accessible to the learning algorithms. The scaling factor can be made time-dependent or state-dependent to implement different types of perturbations.

## DARPA learning introspective control program

FLAIR was deployed across multiple terrains for the DARPA Learning Introspective Control (LINC) program (https://www.darpa.mil/program/learning-introspective-control) that aims at developing learning-based algorithms for ground vehicles, ships, and robotic systems, to respond in real-time to perturbations not taken into consideration at design time.

It included a simulation of the Bernoulli effect (Fig. 2D) using a treadmill illustrated in Fig. 1, a desert sand track, and a chicane track similar to the one used for the experiments reported in this paper.

## Experimental platform and setup

To evaluate FLAIR's capacity to restore operability when faced with unseen perturbations during deployment, we designed 3 circuits that the robot needs to complete, illustrated in Fig. 3. Each circuit tests different types of perturbation described in Section 2.1, in addition to the naturally occurring perturbations, resulting in 5 independently-evaluated sections in total:

- Chicane Static: Following a Chicane with a constant degraded track (Fig. 3A).
- Chicane Dynamic: Following a Chicane with an alternating track degradation over time (Fig. 3A).
- Ramp and Wind: Climbing a slippery ramp with alternating degradation (combination of slippage and damage), followed by a simulated wind, pushing the robot depending on its orientation (Fig. 3B top and bottom).
- Robustness: Climbing a slippery ramp with noise-inducing obstacles that cause vibrations and collisions, complicating the online data collection (Fig. 3C).

We use an automatic driver to remove human factors that may affect evaluation, such as operator fatigue and system experience. In total, we present 740 experiments (20 replications per algorithm and circuit) or ~ 19 h of real-world experiments (excluding preliminary experiments and preparation or reparation time).

## Overcoming real-world perturbations

We consider two metrics to quantify the benefits of FLAIR. First, we measure the Tracking Error: the error between the command sent by the driver and the behaviour that the robot executes, measured by an external VICON motion capture system. It directly reflects operability. Second, we also measure the circuit Completion Time in seconds. We provide additional details on our metrics in the Supplementary Results. We report $p$ values computed using the Wilcoxon Rank test.

We run the automatic driver on the three circuits without ("Driver") and with FLAIR ("Driver + FLAIR"). We also run baselines without any perturbation ("Driver [No Perturbation]" and "Driver + FLAIR [No Perturbation]"), which estimate how well the robot should do under full operability. As baselines, we implement an optimal control algorithm, namely Linear Quadratic Regulator (LQR)[35–37] which has proven effective at robust robotic locomotion[38–40], a state-of-the-art adaptive control algorithm, $\mathcal{L}_1$ Adaptive Control, and an online RL method that learns online the residual to apply to the command to compensate for the perturbation[13] (see Supplementary Baselines).

We provide the results on all circuit sections in Fig. 4. In Table 1, we compute the increase of Tracking Error induced by each method with respect to the baseline "Driver [No Perturbation]" (Table 1A). Additionally, we compute the ratio of Tracking Error increase induced by "Driver + FLAIR" and by "Driver" (Table 1B). This metric gives the proportion of operability that is recovered when using FLAIR compared to not using FLAIR.

Overall, our results show that across all circuit sections, FLAIR reduces the Tracking Error by 74.4% and the Completion Time by 72.2%. FLAIR has been designed to recover operability across a variety of tasks by learning the correct model online to compensate for perturbations. FLAIR reduces the error induced by perturbations by half when compared to our $\mathcal{L}_1$ and LQR baselines, demonstrating that it is able to learn the correct model online and correctly reset it. In comparison, $\mathcal{L}_1$ or LQR can, in some cases, reduce the error by $\approx$ 50%, however we argue that this is insufficient in scenarios requiring high manoeuvrability and that any reduction in perturbation is beneficial to the operator. These results show that a learning based method is able to outperform methods such as $\mathcal{L}_1$ and LQR by halving their tracking error.

These results are achieved with an average time to update FLAIR's model under 225ms for new situations, thanks to the optimisation of our learning algorithm for an efficient operation on onboard hardware (See Supplementary Experimental Platform for more details). In the presence of perturbations making the robot inoperable, FLAIR significantly participates in reconstituting operability while showing very limited overheads in the case of no perturbations. We emphasise that these results are obtained from learning online when encountering the perturbations, without any form of task-specific pre-training. The RL baseline fails to learn how to adequately compensate for perturbations, resulting in unsafe behaviour that necessitates halting its operation before completing the circuit, marked with a red cross in Fig. 4 (additional details in Supplementary Baselines). Due to safety concerns, we didn't try our RL baseline on Ramp, Wind and Robustness since we were not able to keep it safe on flat terrain. Our optimal control baseline, LQR, and our adaptive control baseline, $\mathcal{L}_1$, are able to recover some of the completion times by overshooting, but their tracking error suggests that the robot is worse at following direct commands from the driver. This means that the robot is not following the commands

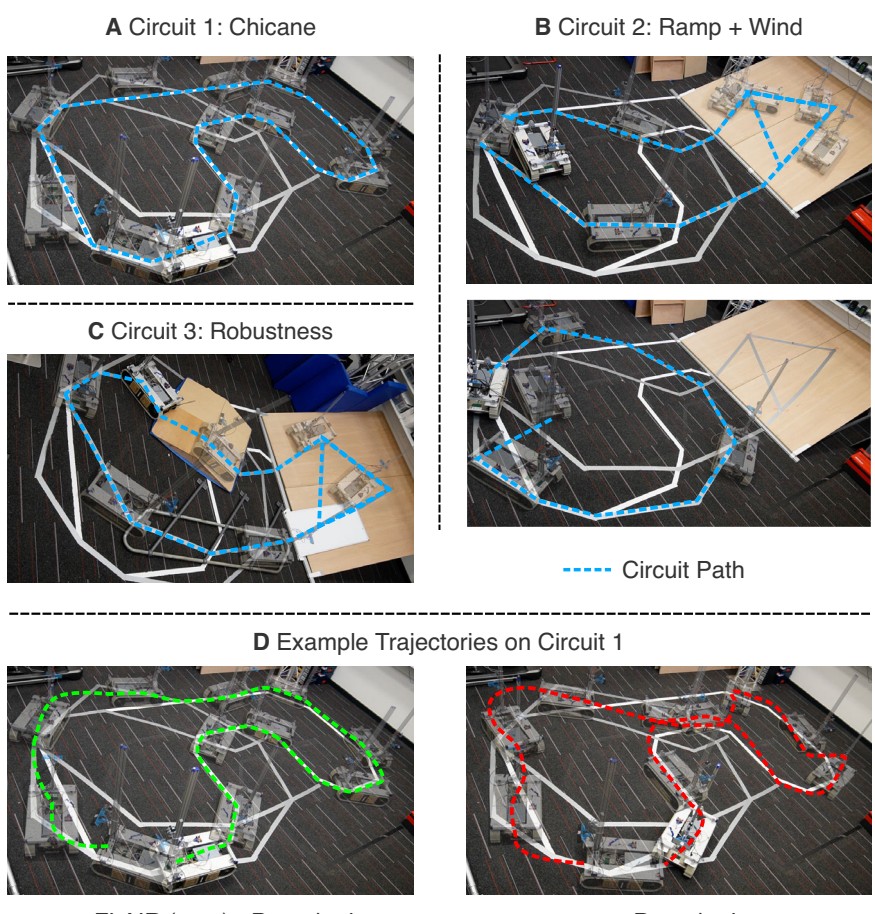

**Fig. 3 | Evaluation circuits. A** Circuit 1 is a chicane run testing either static or dynamic perturbations. **B** Circuit 2 is split into two sections: the Ramp section (top), which combines natural perturbations (slippery ramp) and a static perturbation; and the Wind section (bottom), which tests state-dependent perturbations, in this case a simulated wind. **C** Circuit 3 tests the capabilities of the robot to learn on terrains with additional obstacles that induce vibrations, slippage, and noisy sensor data. **D** The figure depicts two example trajectories on Circuit 1 under dynamic perturbation. FLAIR produces a smoother trajectory (green trajectory, left) and is able to follow and track the circuit path more easily than without adaptation (red trajectory, right).

of the driver as well as FLAIR, making it harder to control. This is especially visible in cases where the damage changes over time, highlighting the time-invariance assumption made by LQR and $\mathcal{L}_1$. We note that the Completion Time can be shorter for a higher Tracking Error, due to the algorithm executing higher velocities than requested. For example, $\mathcal{L}_1$ adaptive control has shorter completion times than LQR but a worse tracking error, which is caused by higher velocity requests, contradicting driver commands (see Supplementary Results).

Looking at each scenario, we can see that for the Chicane Static and Dynamic sections (Circuit 1 in Fig. 3A), driving around the circuit by following tight curves is not easy once the robot is perturbed, especially once the perturbation changes dynamically. FLAIR successfully reduces the Tracking error by 81.1% and 56.6%, outperforming LQR ($p < 1.7 - 05$) and $\mathcal{L}_1$ adaptive control ($p < 2.6e - 05$).

We observe the effect of increasing perturbations on FLAIR's performance by running an ablation of the strength of the perturbation for the Chicane Static section. Fig. 4E shows the evolution of the Tracking Error with increasing strength of perturbations. This result shows that FLAIR maintains a similar operability no matter the strength of the perturbation.

For the Ramp section (Circuit 2 in Fig. 3B), two main challenges had to be overcome: the slippage on the wooden ramp and the additional static perturbation, which makes it harder to make a left turn. As the slippage is only happening occasionally, the combination of these two perturbations does not impair the robot uniformly across the run, forcing FLAIR to constantly adapt to the current effect of the damage. Despite this complexity, FLAIR fully recovers operability on this track (101.0% of tracking error reduction). In comparison, LQR and $\mathcal{L}_1$ adapt to the perturbations but fail to match the performance of FLAIR ($p < 3.7e - 06$ and $p < 7.2e - 07$).

For the Wind section (Circuit 2 in Fig. 3B), FLAIR needs to detect the perturbation change after the Ramp section, and then learn a complex state-dependent perturbation that changes with the robot's orientation. Without FLAIR, the operator performs 40.8% more errors, but once FLAIR is active, the operator is able to come within 10.3% of the baseline performance, reducing the tracking error by 74.4% ($p < 6.3e - 08$). The baselines struggle to follow driver commands accurately, with $\mathcal{L}_1$ failing to execute smooth curves (Fig. 4C), resulting in high Tracking Error.

The Robustness section (Circuit 3 in Fig. 3C) aims to test the robustness of FLAIR in noisy and complex real-world scenarios. In this setup, LQR and $\mathcal{L}_1$ are strong baselines, inducing only a time overhead of 18.9% and 10.3% respectively. However, this setup still impacts the effectiveness of their immediate recovery, as indicated by the error overhead of 23.1% and 25.6% respectively. The results demonstrate that FLAIR successfully ignores the additional confusing signals, inducing smaller error overhead than LQR and $\mathcal{L}_1$, and reducing tracking error by 54.9%.

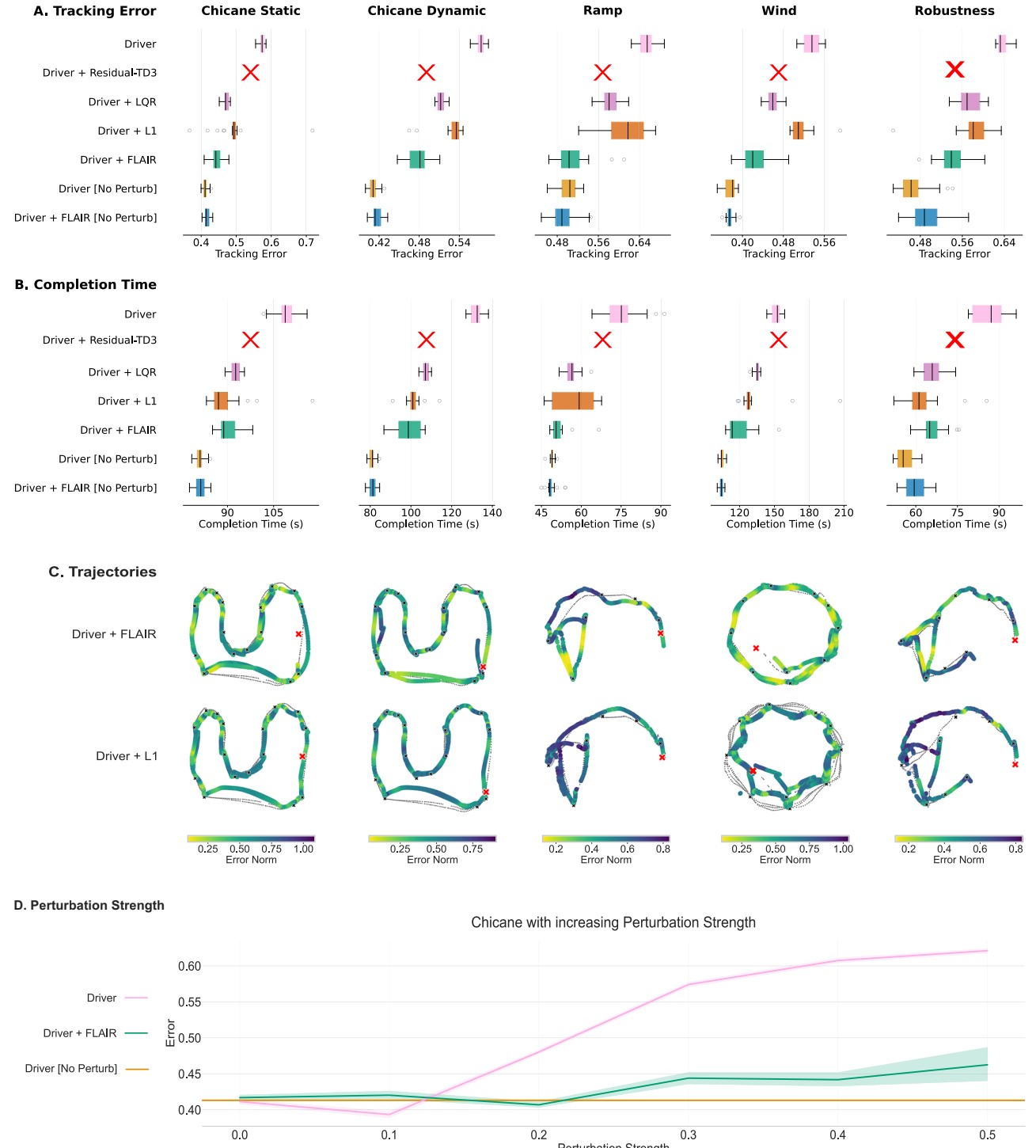

**Fig. 4 | FLAIR's Performance.** We compare FLAIR against baselines to show the benefits on (**A**) Tracking Error and (**B**) circuit Completion Time. In the box-plots, the center line represents the median, the box limits denote the upper and lower quartiles, the whiskers extend to 1.5x the interquartile range, and individual points represent outliers. In the line-plots, the line represents the mean and the error bar the 95% confidence interval. We use a red cross to note the baseline that could not complete a single experimental run. **C** We also display trajectories of one randomly-picked replication of "Driver" and "Driver+FLAIR", using "Driver [No Perturbation]" as reference (black line). The black crosses correspond to the way-points, the red cross to the starting point, and the colour of the trajectory gives the norm of the tracking error between two way-points (the darker the segment, the higher the error). **D** We also run an ablation of the static perturbation on the Chicane to show that FLAIR's performance is robust to increasing perturbation strength.

## Introspection

Another significant strength of FLAIR is its introspection capability to help the operator understand the underlying perturbations. Learning an online model for the current scenario allows for introspection by highlighting how perturbations will impact the robot across the command space and state space. By displaying the function of the perturbation, FLAIR helps the operator understand the situation and choose the best command to avoid states with high perturbations, such as the borders of a canal where the Bernoulli perturbations are the highest. Although various regression techniques (i.e., Kernel

**Table 1 | Summary of FLAIR's Performances. We report the median of the 20 replications**

| Perturbation | | Static | Dynamic | Static | State-Dep. | Static | |
|---|---|---|---|---|---|---|---|
| Circuit | | 1 | 1 | 2 | 2 | 3 | |
| Section | | Chicane Static | Chicane Dynamic | Ramp | Wind | Robust. | Median |
| (A) Tracking Error increase | Driver | 39.6% | 39.3% | 29.8% | 40.8% | 36.8% | 39.3% |
| | Driver + FLAIR | 7.5% | 17.1% | −0.3% | 10.3% | 16.6% | 10.3% |
| | Driver + FLAIR [No Perturb] | 0.7% | 0.7% | −3.1% | −1.5% | 5.5% | 0.7% |
| | LQR | 14.2% | 24.6% | 15.1% | 20.6% | 23.1% | 20.6% |
| | $\mathcal{L}_1$ Adaptive Control | 7.4% | 30.3% | 22.4% | 33.8% | 25.6% | 25.6% |
| (B) FLAIR Tracking Error Reduction | | 81.1% | 56.6% | 101.0% | 74.4% | 54.9% | 74.4% |
| (C) FLAIR Update Time | | 122ms | 123ms | 225ms | 218ms | 225ms | 218ms |

**A** Overhead with respect to the "Driver [No Perturbation]" baseline in terms of circuit Tracking Error. **B** We also quantify the Tracking Error Reduction by FLAIR as the reduction in increased error for "Driver + FLAIR" with respect to the "Driver" (100% means the overhead in time or error is gone). **C** Time needed to update our FLAIR model (ranging from data filtering and model training to the deployment on the robot) in milliseconds for each section. For all metrics, we compare the medians over 20 replications.

regression, Neural Networks, GPs) could be used to learn this function, we prioritize interpretability and transparency, and therefore employ a polynomial model that balances expressive power with ease of visualization and explanation for the operator.

In Fig. 5A–D, we display examples of the diagnosis produced by FLAIR for the perturbations considered in our experiments. Our system proposes an approximated model of how the perturbations impact the robot with respect to the relevant dimension of the state $s$ using a Taylor expansion of the form $d'(s) = (\alpha + \beta s + \gamma s^2 + \delta s^3)$ (see Methods - Global Adaptation to Perturbations for more information). For example, in the case of the Bernoulli perturbation in a canal, $s$ is the $y$-displacement of the robot, while in the case of a Wind perturbation, $s$ is the yaw orientation of the robot with respect to the wind direction.

For the static perturbation (Fig. 5A), we display an example on the Ramp section (Circuit 2 in Fig. 3B) where the perturbation factor is a combination of static perturbations and slippage. FLAIR is first able to find the corresponding polynomial with a constant $\alpha$ term. Then, once the robot starts slipping on the ramp, FLAIR quickly adjusts and proposes a new polynomial with a higher constant value that accounts for both the static perturbation and the slippage.

Fig. 5B shows an example of dynamic perturbations introspection on the Chicane section (Circuit 1 in Fig. 3A). FLAIR adjusts to the four changes in perturbations by learning a new model and predicting the correct corresponding $\alpha$ term each time. This shows that FLAIR rapidly learns a new model every time in a short amount of time (short ramp-up before plateauing at the fixed rate).

In Figure 5C and Fig. 5E, we display the introspection for the state-dependent wind perturbation (Circuit 2 in Fig. 3B). This perturbation pushes the robot to the side with a force relative to the yaw orientation. In the upper plot of Fig. 5C, we show that FLAIR learns a linear polynomial (i.e., $\beta = -0.159$ and $\alpha = \gamma = \delta = 0$) that best approximates the perturbation's effect. Since the wind changes with the state of the robot, FLAIR needs to adjust the commands at each different orientation angle of the robot. We visualise this by plotting the map showing the nominal commands and their predicted behaviours for multiple states (i.e., orientation angles) in Fig. 5E.

As part of the LINC program, FLAIR was tested on a task including a treadmill to simulate a narrow water channel, exposing the robot to Bernoulli forces (https://youtu.be/D8Wbi_lcN4I?si=MJGsm1VqirR2_3nQ). We include this scenario in Fig. 5D, F, as well as in our Supplementary Movie 1, to show the perturbations predicted by FLAIR in the case of state-dependent Bernoulli perturbation. FLAIR successfully predicts a second-order polynomial dependent on the $y$-position of the robot on the treadmill, approximating the expression of a Bernoulli perturbation (i.e., $\gamma \approx 3.2$) which is exponential in nature.

### Generalisation to other robot morphologies

To further illustrate the performance of FLAIR, we similarly apply it to a simulated hexapod robot with 18-degrees-of-freedom per leg. On this robot, we pre-trained a low-level controller that allows the robot to move in every direction. We then use FLAIR hierarchically to adapt to a damaged leg (i.e., weakened actuators) and follow a chicane circuit. FLAIR reduces the tracking error of the driver alone by 138.8%, outperforming it. This result showcases that FLAIR's hierarchical approach can be applied to diverse types of robots. The detailed results are in Supplementary Hexapod Results.

### Discussion

The presented results showcase the capabilities of our proposed approach, FLAIR, to overcome perturbations that naturally occur when operating locomotion systems in real-world scenarios. Before our work, overcoming such perturbations required re-designing new controllers to handle new perturbations encountered either by including them in a simulator or modelling them as part of a model-based system. FLAIR demonstrates that anticipating and accounting for all types of perturbations might not always be necessary. By taking a principled approach in designing high-level controllers, we show that it is possible to augment the existing base controller to function as if no perturbation were present. We demonstrate with FLAIR that such high-level controllers can successfully learn and adapt fully online on an embedded system without requiring any task-specific pre-training or use of external compute resources or sensors. Our experiments demonstrate the performance and robustness of FLAIR in enabling the recovery of operability across a diversity of tasks and perturbations at deployment time in only a few seconds. They also highlight the ability of this method to quickly identify new perturbations and reduce their impact. Additionally, the benefit of FLAIR goes beyond improved operability and provides introspection capabilities that help the operator understand the current situation and make informed decisions.

A direction for future work is to further deploy FLAIR on other vehicles and robotic systems. Exploring different types of systems, such as non-wheeled or even flying systems, could open the door to new types of applications and new types of perturbations that were not considered in this work. More generally, a major opportunity for future work is to use the proposed approach as a starting point in the development of a general-purpose driving assistance module that could be deployed on any locomotion system. Such a module could be added as an external component to an existing system and would augment it and enforce its robustness to perturbations. Finally, as introspection is one of the major benefits of the current FLAIR

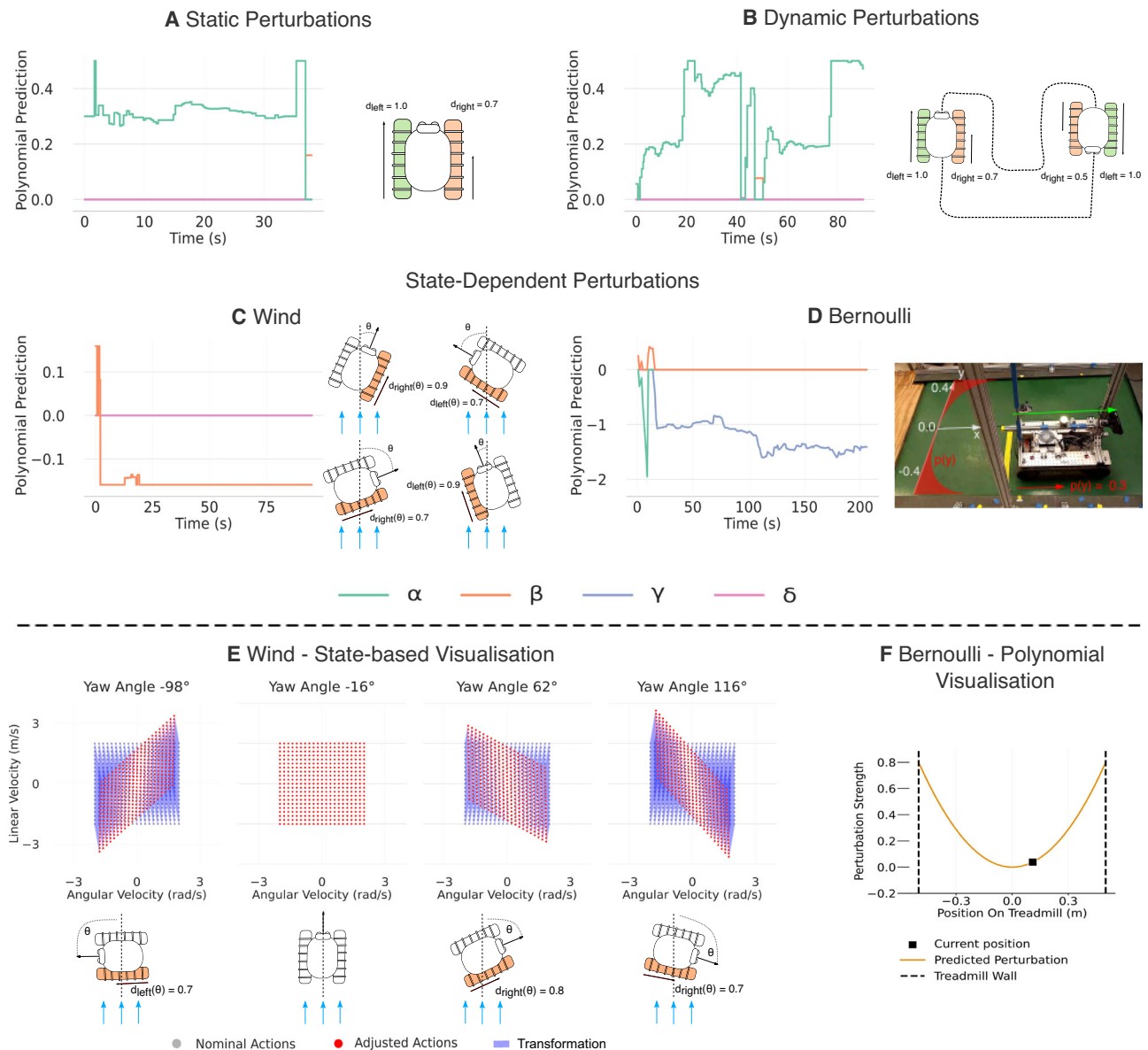

**Fig. 5 | Introspection abilities.** FLAIR is learning an internal model of the effects of perturbations by approximating them with a Taylor expansion. Lines represent a single random replication for each perturbation, selected from $n = 20$ independent experiments. **A**–**D** We display the polynomial prediction ($\alpha + \beta s + \gamma s^2 + \delta s^3$) for the relevant dimension $s$ of the state, made by FLAIR for each type of perturbation considered in the experiments. **E** For the Wind perturbation, which is state-dependent with $s$ being the yaw orientation of the robot with respect to the wind direction, we display how the command maps derived from the perturbation model are adjusted based on the state of the robot. **F** For the Bernoulli perturbation, which is state-dependent with $s$ being the $y$-displacement of the robot, we display a visualisation of the perturbation strength with respect to robot state derived from the perturbation model.

system, it could both assist operators in driving and also enable constant system diagnosis.

While our experiments with FLAIR cover four sets of diverse scenarios, it is not possible to extrapolate these performances to every type of perturbation. For example, a change in weight distribution or friction that requires more force to compensate than the tracks can produce will make the perturbation unrecoverable. More generally, FLAIR cannot fully recover operability when the force required to counteract a perturbation exceeds the maximal strength of the actuators. We study this in the Supplementary Boundary Analysis. Additionally, we tested FLAIR on actuator-channel perturbations for reproducibility and accurate evaluation. While FLAIR was also evaluated on true (non-actuation-based) disturbances as part of the LINC program, true unmatched disturbances (e.g., lateral forces independent of actuation) are out of scope for this first version.

Additionally, FLAIR relies on the sensory data from onboard sensors to learn online. In the case of complete sensor failure, wrong reading due to disturbances (e.g., magnetic fields), or extreme noise on the sensor signal, our method will fail. FLAIR was designed to mitigate sensor noise and incorrect readings through its dataset creation process and by aggregating information across multiple datapoints rather than relying on single predictions. However, a potential future work would be to combine methods from fault-tolerant control with FLAIR to overcome these shortcomings.

Although our results show that FLAIR generalises to wheeled vehicles and legged robots, this is a first approach that is built on the assumption of differential drive dynamics. This is due to FLAIR's global adaptation mechanism, where perturbations are modelled as a velocity scaling function (see Supplementary Boundary Analysis). For more complex robots, such as Humanoids or Robotic Arms, this assumption

is violated, and perturbations may quickly lead to unrecoverable states (e.g., falling), making online learning with this first version of FLAIR challenging. A promising line of work opening up is the learning of a more specialised global adaptation function (e.g., via Deep Learning) for the hierarchical architecture of FLAIR to accommodate more complex robots.

## Methods

### Hierarchical online adaptation

The presented method, FLAIR, is a hierarchical controller that reconstitutes the operability of a robot under various perturbations (see Algorithm 1). FLAIR receives the command sent by the operator and adjusts this incoming command before sending it to the low-level controller. This adjustment ensures that the behaviour induced by the command matches the one desired by the operator. The hierarchical architecture allows FLAIR to abstract the search space[24,41–43] by directly reusing good solutions and quickly adapting to new scenarios.

Figure 2E gives an overview of the FLAIR pipeline. To capture the behaviour of the robot, FLAIR uses an asynchronous approach to collect the onboard sensory data corresponding to the robot's response (i.e., the Dataset Creation process). FLAIR then matches the commands and behaviours before using them as a dataset for its internal model $f$ to produce a more accurate command selection at the next incoming command. FLAIR learns a command-behaviours mapping, which is a transform $f(\mathbf{a}, \mathbf{s})$ to map a command vector $\mathbf{a} \in \mathcal{A}$ and its associated behaviour vector $\mathbf{b} \in \mathcal{B}$ to the new behaviour space $\mathcal{B}'$ with a high-level mapping:

$$f : \mathcal{A} \times \mathcal{S} \to \mathcal{B}' \qquad (2)$$

Once FLAIR has mapped the commands to the new behaviours, it selects the new command $\mathbf{a}'$ which has the closest behaviour under perturbations to the desired behaviour $\mathbf{b}$:

$$\pi(\mathbf{s}, \mathbf{b}) = \arg\min_{\mathbf{a}'} \left\| \mathbf{D}(f(\mathbf{a}', \mathbf{s}) - \mathbf{b}) \right\|$$
$$\text{subject to} \ -2 < \mathbf{a}' < 2 \qquad (3)$$

$\mathbf{D}$ is a diagonal matrix with per-dimension scaling factors. In our case, we use the Identity matrix. In the following, we refer to the robot's controller as the low-level controller $\pi_{\text{low}}$ and to FLAIR as the high-level controller $\pi$. Using its internal model $f$, FLAIR then produces new commands $\pi(\mathbf{s}, \mathbf{b}) = \mathbf{a}'$ which are then fed into the low-level controller. In the unperturbed case, the model $f$ is an identity function of the command and behaviour, as the executed command directly yields the desired behaviour. To speed up the optimisation in Equation (3), we uniformly discretize the command space by a grid of size $101 \times 101$ ranging from -2 to 2 for both linear and angular velocity. 101 was chosen as a trade-off between the precision of 51 points and the computational cost of 151 points. The function $f$ is then evaluated over all grid points to identify the optimal action. This brute-force optimisation of 10, 201 commands is done online and in parallel on the GPU due to the optimised software.

Adapting to perturbations can be decomposed into two steps. First, we want to capture general changes to the system (e.g., a flat tyre), but perturbations rarely impact the entire command space uniformly. For instance, a loss of traction can happen only when a command exceeds a certain threshold. Thus, secondly, in addition to the adaptation to the global changes, FLAIR learns how local perturbations affect individual commands. We detail these steps in the following sections.

FLAIR uses a Gaussian process (GP)[44,45] to learn $f$ and model the non-linear effects of the perturbations online. GPs are a family of stochastic processes that are particularly attractive for this kind of regression problem because of their data-efficiency and uncertainty quantification. These characteristics make them an ideal choice to model the changes in behaviours caused by a change in dynamics[46].

Although GPs are data-efficient, they rely on smoothness assumptions of the learned function, which can be violated when encountering noisy datapoints, caused by collisions, sensory failures, or sudden ground failures. By leveraging a Soft Outlier Rejection mechanism, we are able to filter out highly variable data points that could hinder the learning of the GP. We demonstrate the effectiveness of this mechanism in the Robustness circuit, showing that FLAIR can filter out noisy datapoints and keep a strong level of robustness. We note that FLAIR does not actively learn non-stationary functions but rather relies on resetting mechanisms and filtering of training datapoints. We leave the exploration of including models that can learn non-stationary functions to future work.

**Algorithm 1.** FLAIR: Hierarchical Control with Global and Local Adaptation

1: **Require:** Low-level controller $\pi_{low}$, desired behaviour $b$ from driver, state $\mathbf{s}$
2: **while** Operating Robot **do**
3: Hierarchical Control:
4: Get command $\mathbf{a}$ for desired behaviour $\mathbf{b}$ using learned transform $f$
5: $\mathbf{a} \leftarrow \arg\min_{\mathbf{a}'} \|f(\mathbf{a}', \mathbf{s}) - \mathbf{b}\|$
6: Execute $\pi_{low}(\mathbf{a})$ on robot
7: Data Collection:
8: Observe behaviour $\mathbf{b}_{\text{obs}}$
9: Reduce observation noise and match actions with behaviours
10: Update dataset to maximise command-space coverage and filter outliers
11: Learning Adaptation:
12: Compute behaviour error: $e = |\mathbf{b} - \mathbf{b}_{\text{obs}}|$
13: Update global model of $f(\mathbf{a}, \mathbf{s})$ based on $e$
14: Update local model of $f(\mathbf{a}, \mathbf{s})$ based on $e$

### Operability of the robot

Similarly to previous works[20,24,47,48], we make an important distinction between commands and behaviours. Commands $\mathbf{a}$ are the input given to $\pi_{\text{low}}$ and the behaviours $\mathbf{b}$ are observed characteristics (e.g., speed of the robot) caused by the execution of $\mathbf{a}$ by the robotic system. For the robotic system we consider, the commands $\mathbf{a}$ are the desired linear velocity $v_x$ and the angular velocity $\omega_z$ (i.e., $\mathbf{a} = \begin{bmatrix} v_x & \omega_z \end{bmatrix}^T$) of therobot and the behaviours $\mathbf{b}$ are the observed velocities measured by the sensors.

We consider that a robot is operable when the behaviour $\mathbf{b}$ induced by the command $\mathbf{a}$ is the same as the behaviour $\mathbf{b}$ induced by the command $\mathbf{a}$ under the unperturbed conditions considered when the controller was designed. Thus, the robotic system considered in this work is operable when the requested velocities (sent as commands) correspond to the observed velocities. The role of FLAIR is to recover the operability of a robot once its behaviours do not correspond to the desired commands anymore due to perturbations.

**Global adaptation to perturbations.** To model the effect of perturbations on the operability of the robot, FLAIR uses a two-stage approach. First, FLAIR learns how the perturbations impact the command-behaviour mapping across the full command space, which we refer to as "Global Adaptation". Second, to model localised issues such as sticky points, FLAIR learns how individual commands differ in behaviour, which we name "Local Adaptation".

To capture system-wide perturbations, FLAIR parametrises these directly into the kinematics model of the differential-drive vehicle. This enables quick discovery of the global change in the robot's behaviours with respect to the perturbations for every command. Given that our system is a differential drive vehicle, the main ways the system

interacts with the world are via the two tracks and the ground. Therefore, most of the situations, such as impairments, loss of friction, or uneven terrains, can be represented by an under- or over-actuation of the tracks' interactions with the ground. To this end, we denote the parameters for the external perturbations on each track as $d_{\text{left}}$ and $d_{\text{right}}$ and their estimation by FLAIR as $d'_{left}$ and $d'_{right}$. Both values are approximations of how the external perturbations scale the robot's capabilities on either the left or right tracks of the robot (e.g., the traction). We denote the left and right velocities of the tracks by $v_l$ and $v_r$ and the width of the robot by $h$, which gives us the following standard differential drive kinematic model:

$$\begin{bmatrix} v_l \\ v_r \end{bmatrix} = \begin{bmatrix} v_x + \frac{\omega_z}{2} h \\ v_x - \frac{\omega_z}{2} h \end{bmatrix} \tag{4}$$

and

$$\begin{bmatrix} v_x \\ \omega_z \end{bmatrix} = \begin{bmatrix} \frac{v_l + v_r}{2} \\ \frac{v_r - v_l}{h} \end{bmatrix} \tag{5}$$

Under perturbations, we assume that $v_l$ or $v_r$ is scaled by $d'_{left}$ or $d'_{right}$, resulting in a new velocity $v'_x$ and $\omega'_z$. By scaling $v_l$ or $v_r$ and substituting Equation (4) into Equation (5), we get the equation for the new velocity $v'_x$ and $\omega'_z$ under perturbations as a matrix multiplication:

$$\begin{bmatrix} v'_x \\ \omega'_z \end{bmatrix} = \begin{bmatrix} \frac{d'_{left} + d'_{right}}{2} & \frac{d'_{right} - d'_{left}}{4} h \\ \frac{d'_{right} - d'_{left}}{h} & \frac{d'_{left} + d'_{right}}{2} \end{bmatrix} \begin{bmatrix} v_x \\ \omega_z \end{bmatrix} \tag{6}$$

with $d'_{left}, d'_{right} \in [0, 1]$.

If $d'_{left} = 1$ and $d'_{right} = 1$ then our robot is not impacted, however when $d'_{left} = 0.5$ and $d'_{right} = 1$, then our robot can only apply 50% of its forces on the left track. These perturbations can be caused by motor damage (e.g., the motor of one track is not able to run at full capacity), constant wind forces from either side, or slippage (i.e., the robot cannot apply all its force onto the ground, resulting in weaker movements). In scenarios where only one side of the robot is perturbed at a time, we can replace $d'_{left}$ and $d'_{right}$ with $d' = (d'_{right} - d'_{left})$. This forces either $d'_{left}$ or $d'_{right}$ to be set to 1. In this condition we can rewrite Equation (6) as a linear equation:

$$\begin{bmatrix} v'_x \\ \omega'_z \end{bmatrix} = (d' \begin{bmatrix} \frac{\alpha}{2} & \frac{h}{4} \\ \frac{1}{h} & \frac{\alpha}{2} \end{bmatrix} + I) \begin{bmatrix} v_x \\ \omega_z \end{bmatrix} \text{ with } \alpha = \begin{cases} 1, & \text{if } d'_{left} = 1. \\ -1, & \text{if } d'_{right} = 1. \end{cases} \tag{7}$$

Since perturbations are not constant but may vary with the state, $d'$ can be a function of the relevant state dimension $s$. In our case, we assume that an engineer has selected the relevant state information $s$, such as the displacement or the rotation of the robot, depending on the perturbations. Here, we use the rotation across all our experiments (except the treadmill). We approximate the global command-behaviour mapping $d'(s)$ with a Taylor expansion up to the third degree with parameters $\alpha, \beta, \gamma, \delta$. We decided to use a third-degree polynomial since the Bernoulli force used during the LINC experiments is quadratic, and to cover any complex non-linear perturbations (e.g., complex drag), we add a third degree to the polynomial that can be used to approximate perturbations. The function could be approximated with other methods (e.g., Deep Learning) or another Taylor series. This could be useful to integrate higher-dimensional state information into the function $d'(\mathbf{s})$, but in our experiments, we did not explore the function choice further, given the positive results with the Taylor series.

$$\begin{bmatrix} v'_x \\ \omega'_z \end{bmatrix} = (d'(s) \begin{bmatrix} \frac{-\text{sign}(s)}{2} & \frac{h}{4} \\ \frac{1}{h} & \frac{-\text{sign}(s)}{2} \end{bmatrix} + I) \begin{bmatrix} v_x \\ \omega_z \end{bmatrix} \tag{8}$$

with

$$d'(s) = (\alpha + \beta s + \gamma s^2 + \delta s^3)$$

and

$$\text{sign}(x) = \begin{cases} 1, & \text{if } x >= 0. \\ -1, & \text{if } x < 0. \end{cases}$$

Equation (8) can be solved with the least squares method[49] to find $d'$, which is directly implemented on GPU. This allows FLAIR to rapidly adapt to different perturbations once the data is collected and quickly react to perturbations that impact the overall command space of the robot.

**Local adaptation to perturbations.** For perturbations that only affect certain commands and not the full command space, FLAIR additionally captures how certain perturbations can impact localised commands.

Specifically in our case, $f$ is trained on desired commands $\mathbf{a} = (v_x, \omega_z)$ and corresponding behaviours $\mathbf{b}$. The new behaviour $\mathbf{b}' = (v'_x, \omega'_z)$ for a new command $\mathbf{a}'$ is then the predictive mean $\bar{\mathbf{f}}$ of the following GP:

$$f(\mathbf{s}', \mathbf{a}') \sim GP(m(\mathbf{s}', \mathbf{a}'), k((\mathbf{s}, \mathbf{a}), (\mathbf{s}', \mathbf{a}')))$$

with the predictive mean:

$$\bar{\mathbf{f}} = m(\mathbf{s}', \mathbf{a}') + K((\mathbf{s}', \mathbf{a}'), (\mathbf{s}, \mathbf{a})) K_y^{-1} (\mathbf{b} - m(\mathbf{s}, \mathbf{a}))$$

To combine both the local and global adaptation to perturbations, FLAIR uses the Equation (8) as mean function $m(\mathbf{s}, \mathbf{a})$ to our GP. By optimising Equation (8), FLAIR can update its beliefs around the global impact of the perturbations while adjusting for local disparities with the GP model. The polynomial parameters in Equation (8) are only solved via the least squares method, and then used as fixed GP mean.

The kernel for the GP used across this paper is the Radial Basis Function Kernel[50] with different length-scales per input dimension, which are the commands $\mathbf{a}$ and the state parameter $\mathbf{s}$ (see Supplementary Methods). The length-scales, noise, and variance parameters of the kernel and GP are considered robot-wide parameters. They are tuned by optimising the log-marginal-likelihood using 1000 data-points collected while the robot operates under non-perturbed conditions. After these few seconds of initial operation, we freeze the parameters throughout all the experiments. Different hyper-parameters have been tried (e.g., learning them on different terrains) with little difference in adaptation performance. We did not experiment with alternative kernels as the RBF kernel performed very well in our experiments. The Soft Outlier Rejection mechanism likely removes a lot of the non-smooth data-points which probably helps the GP learn well with a RBF kernel. Other kernels might be better suited for specific use cases or robot types, but we leave this investigation for future work, as it falls outside the scope of the present study.

Our local and global adaptations fit a continuous function to approximate the impact of the damage. This can be extended to discontinuous functions in case the perturbation functions are more complex by using different global adaptation functions.

FLAIR trains the GP from scratch every time, no task-specific pre-training prior to deployment is done. For efficiency purposes, we limit the full dataset array to 1000 points to train the GP used in our experiment. More details on the dataset management can be found in following sections. We parallelised and optimised our code to reduce the total learning and execution time on our on-device GPU (see Supplementary Experimental Platform) to less than 225 ms which includes the learning of the GP, data filtering, command updates and the least-squares optimisation for the polynomial parameters in Equation (8).

The hyper-parameter optimisation only happens once during the first few seconds of operation, and is not re-run at any point during the deployment of FLAIR. Even after a reset, the same hyper-parameters are used, and the only learning that happens is the local (GP) and global (Least Squares) optimisation.

## Continuous scenario identification

During deployment, the situation encountered by the robot can change abruptly: sudden failures of components, shifts in terrain, changes in weather conditions, etc. A major problem in sudden changes of scenarios is the shift in data distribution that the system has to deal with[51,52]: when a change occurs, FLAIR has learned a model $f$ to adapt to an old scenario and might not immediately reflect the new perturbation given the collected data from the old scenario. To learn its model $f$, FLAIR stores data in a dataset that acts as its memory (detailed in the Section - Dataset Creation for Robust Online Learning). FLAIR is able to forget the old scenario if enough time passes and new data points slowly take over the buffer. However, this replacement is slow, and in the meantime, the new perturbations might have a significant impact on the operability of the controller in use. Thus, in addition to its dataset, FLAIR also uses a Continuous Scenario Identification module, which enables quick and efficient detection of drastic changes in perturbations. Once such changes are detected, FLAIR resets its memory by emptying the dataset to learn a new model $f$ that accounts for the new perturbation in seconds, which also includes the optimization of the parameters in Equation (8).

FLAIR relies on sensory information from the robot to detect changes in perturbations that could possibly affect the learned model. To measure how much the predictions from our internal model $f$ deviate from the actual effect of the perturbations, FLAIR first calculates what the model would have predicted as behaviour $\mathbf{b}'$ given a desired command $\mathbf{a}'$. We assume that the unperturbed system has a behaviour $\mathbf{b}$ for a command $\mathbf{a}$ that corresponds to the command itself (i.e., the velocity of the robot). With this, FLAIR calculates the predicted intent error vector $\iota$ between desired behaviour $\mathbf{b}$ by the operator and the predicted behaviour $\mathbf{b}'$ by the model to measure what change in behaviour our model predicts given the perturbations:

$$\iota = |\mathbf{b} - \mathbf{b}'| \tag{9}$$

In addition, FLAIR calculates the observed execution error vector $\boldsymbol{\epsilon}$ between the non-adjusted command $\mathbf{a}$ (i.e., command that the operator sends to the robot and is not adapted by FLAIR) and its associated behaviour $\mathbf{b}$ that is expected, and the observed behaviour $\mathbf{b}'_{obs}$ recorded by the sensors:

$$\boldsymbol{\epsilon} = |\mathbf{b}'_{obs} - \mathbf{b}| \tag{10}$$

Both error vectors are normalised by a characteristic behaviour scale $\boldsymbol{\sigma}_b = \begin{bmatrix} 1 & 1.25 \end{bmatrix}$ in our case:

$$\tilde{\iota} = \frac{|\mathbf{b} - \mathbf{b}'|}{\boldsymbol{\sigma}_b}, \tilde{\boldsymbol{\epsilon}} = \frac{|\mathbf{b}'_{obs} - \mathbf{b}|}{\boldsymbol{\sigma}_b}.$$

Finally, FLAIR subtracts both errors to measure how much the reality (i.e., the system) deviates from what the model would have predicted, which we call model agreement $\upsilon$:

$$\upsilon = ||\tilde{\iota} - \tilde{\boldsymbol{\epsilon}}||_2 \tag{11}$$

Once the model $f$ starts disagreeing with the reality measured by the sensors, $\upsilon$ increases, and FLAIR triggers a reset of the dataset by discarding any data point collected so far and starts re-learning the new scenario. In our case, we chose a threshold of $k = 0.33$ to trigger this reset. Note that in the case of a perturbation with only a mild

impact that fails to reach this threshold, the dataset management mechanisms in FLAIR, described in the next sections, would still ensure that this new perturbation eventually becomes predominant in the replay buffer, thereby allowing the model update to handle it as if it had undergone a reset. If the robot is not exposed to any perturbations, the execution error $\boldsymbol{\epsilon}$ is close to 0 since a command $\mathbf{a}$ produce the expected behaviour $\mathbf{b}$ and the intent error $\iota$ tells how well our model has identified that there is no perturbation (being 0 when $\mathbf{b}' = \mathbf{b}$). In the case of a perfect learned model, $\iota$ should be equal to $\boldsymbol{\epsilon}$, showing that the model has learned the impact of the perturbations on the behaviours because $\mathbf{b}'_{obs} = \mathbf{b}'$ and the executed command $\mathbf{a}'$ should produce a behaviour $\mathbf{b}'_{obs}$ that equals the desired behaviour $\mathbf{b}$ (i.e., requested command before going through our system).

## Dataset creation for robust online learning

FLAIR learns $f$ online from the onboard sensors data. Processing and storing data is crucial to guarantee the accuracy of the perturbation model and the efficiency of our approach. The Dataset Creation process updates and maintains the dataset that is used to learn the model $f$. It can be divided into three main parts, detailed below. First, the Data Point Creation centralises the data coming from the sensors and the controller to build data points that accurately report the mapping from command to behaviour to guarantee the exactness of the model. Second, the Command-aware Data Storage continuously maintains and orders incoming data points into a dataset. Third, the Soft Outlier Rejection detects incorrect or noisy data points that might confuse the model and dynamically hides them from the modelling process to increase robustness.

**Data point creation.** Providing model training with a high-quality dataset first requires building the data points from raw data from the sensors and the controller to accurately report the effect of the current system dynamics and correctly map the controller's commands to their induced behaviours.

A given command sent to a dynamical system requires time to propagate in a resulting sensor reading (i.e., behaviour) due to the system dynamics and inertia. Thus, commands from the controller and behaviour from the sensors are first dynamically synchronised using cross-correlation[53,54].

The behaviour readings from the onboard sensor have a higher frequency than the command signal. Thus, once the command and sensor signals are synchronised, the second step involves selecting a data point that best represents the real impact of a command from the multiple matching behaviour points. FLAIR assumes the more optimistic scenario, i.e., the scenario where the impairment to the system is minimal. This is an intentional safety-oriented heuristic that guarantees conservative adaptation by favouring data points with lower command-behaviour error. To reduce the impact of noise, the induced behaviour $\mathbf{b}$ is selected as the inter-quantiles closest (inspired by the inter-quantiles mean, also known as trimmed mean[55]): the closest to the command $\mathbf{a}$ after removing the first and last quantiles. This step returns a final data point, composed of a command and its induced behaviour.

**Command-aware Buffer.** For learning to be fast and efficient, it is key to maintain a reasonable number of data points in the dataset. A straightforward way to do this would be to use a first-in-first-out (FIFO) buffer architecture, where older data points are replaced with newer ones. However, to ensure an accurate model of the perturbation, it is key to maintain data that covers the command space well. Data from rare commands contains important information that should not be deleted in favour of over-represented recent commands (e.g., going forward). A simple FIFO buffer might lead to such forgetting and prevent maintaining good coverage of the command space. To overcome this issue, we propose discretising the command space into a

grid (inspired by ref. 56). In this grid, the cells are independent FIFO buffers, all with the same size, referred to as their depth. When a new data point enters the grid, it replaces the oldest data points in its corresponding cell of the command space. Importantly, this discretisation is used solely to structure the FIFO architecture; the data themselves remain continuous and are not discretised. This is coupled with a global FIFO mechanism that enforces a total size by removing the oldest data points in the grid, regardless of their cells. We refer to this double-FIFO as a Command-aware Buffer. A FIFO Buffer and a Command-aware Buffer of the same size may contain the same number of data points, but the coverage of the command space of the Command-aware Buffer would be better. This property ensures a better dataset for learning the perturbation's effect model $f$.

**Soft outlier rejection.** FLAIR requires small amounts of data to learn a perturbation's effect model. While this data-efficiency is a strength for real-world deployment, it also raises one major issue: a single noisy or incorrect data point is enough to heavily confuse the system, and lead to wrong predictions. To ensure robustness in FLAIR, we propose a last step in constituting the dataset: the Soft Outlier Rejection. This mechanism detects data points that appear to be outliers in the current dataset and temporarily hides them from the model learning process. Importantly, our mechanism does not delete those data points, it only caches them (hence the "Soft" in the name). Data points that appear to be outliers at the current time might just be the first occurrences of incoming global changes induced by a new perturbation. Maintaining them in the dataset thus allows potential changes to incrementally take over and become the new norm if more of such data is observed, which increases the certainty. On the contrary, deleting these data points might prevent the model from seeing dynamic changes in the environment or in the perturbations. The Soft Outlier Rejection is re-run each time a batch of data points is added to the dataset, thus allowing to dynamically select which points should be hidden whenever the dataset is modified.

The Soft Outlier Rejection relies on the grid structure of the Command-aware Buffer described in the previous subsection. Each cell of the grid contains multiple data points that correspond to the same command. Outliers are extracted from these data points based on their behaviour-space distance to the other data points in the same cell, referred to as their novelty[57]:

$$\mathrm{nov}(i) = \sum_{k \neq i \text{ in same cell}} ||\mathbf{b_k} - \mathbf{b_i}||^2$$

If one data point has high novelty, meaning it leads to a behaviour drastically different from all the other data points with similar commands, it is considered potentially inaccurate and thus is hidden from the model learning process. Thus, the Soft Outlier Rejection acts as a sieve that limits the impact of noise and inaccurate data on the system and enforces robustness.

## Data availability
All data needed to evaluate the conclusions in the paper are present in the paper or the Supplementary Information. The collected experimental data is available via Zenodo (https://doi.org/10.5281/zenodo.14546657).

## Code availability
We make our code available publicly available to facilitate later work at (https://github.com/adaptive-intelligent-robotics/FLAIR) archived at via Zenodo (https://doi.org/10.5281/zenodo.18336476[58]).

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

## Acknowledgements

We want to thank the members of the Adaptive and Intelligent Robotics Lab, the members of our performer team at the LINC challenge, namely members of Peraton Labs and the members of the Safe Robotics Laboratory from Princeton University, and Stefan Leutenegger for the fruitful discussions and their feedback. Additionally, we thank the Sandia National Labs Team for their help with the onboard computer setup. Funding: This work is funded in part by the Defense Advanced Research Projects Agency (DARPA) Learning Introspective Control (LINC) program.

## Author contributions

A.C. directed the research activities. M.A. and A.C. designed and implemented the initial version of the FLAIR architecture. All authors: M.A., M.F., B.L., and A.C. contributed to the development of all of the modules of FLAIR. M.A. had a particular focus on the Hierarchical Online Adaptation module and developed its core functionalities. M. F. focused on the Dataset Creation module and developed its core functionalities. M.A. and M.F. designed the experimental setup, implemented the baselines, designed the simulated experiments, performed the real-world experiments and processed the results. All authors: M.A., M.F., B.L., and A.C. analysed the results, discussed improvements and wrote the manuscript.

## Competing interests

The authors declare no competing interests.
