## [Transparent Peer Review file · Nature Communications]

Getting Robots Back On Track by Reconstituting Control in Unexpected Situations with Online Learning

Corresponding Author: Dr Maxime Allard

Version 0:

Reviewer comments:

Reviewer #1

(Remarks to the Author)

This work introduces FLAIR, an online hierarchical adaptation method that adjusts robot commands to recover operability in the presence of perturbations. The method combines global linear adaptation and local residual modeling using Gaussian Processes. The results are promising, showing significant recovery across various perturbation types. I reviewed an earlier version and greatly appreciate the efforts the authors made to improve this manuscript.

While the overall framework is clear and experimentally grounded, I have five primary concerns that I believe are essential to clarify before publication:

1. The core assumption in FLAIR is that global perturbations can be captured with a low-degree polynomial (Eq. 7), and local perturbations can be approximated with a GP using 1000 data points. While this is effective for the experimental circuits, real-world disturbances are often highly nonstationary, rapidly varying, or discontinuous (e.g., sudden ground loss, wind bursts, partial sensor dropout).

Such behaviors may violate the smoothness assumptions in both Taylor approximations and GPs, and potentially cause misleading generalizations. I am concerned that in these cases, online modeling could deteriorate performance, especially under rapid perturbation shifts that are too subtle for the model reset condition to trigger.

2. The method defines a model agreement threshold $u > 0.33$ to reset and relearn a new model. However, if the real-world perturbation causes moderate but persistent disagreement, just below the threshold, the system may continue using an outdated model, resulting in compounding control errors.

There is no theoretical or empirical analysis of how sensitive the system is to sub-threshold dynamics, nor whether this can lead to performance being worse than non-adaptive baselines in some conditions. This safety-critical edge case deserves attention.

3. Although the results demonstrate solid performance, it's not obvious that FLAIR's performance is decisively better than simpler baselines in all cases. In several cases, L1 or LQR recover a substantial fraction of functionality (often >50%) without any learned model. These methods are intrinsically robust due to the control structure, while FLAIR depends on the correctness of learned models and online resets.

4. In the RL literature, several works (e.g., Residual RL, Adaptive RL with online fine-tuning, Meta-RL) handle nonstationarity by combining offline policies with online correction or adaptation layers. These share similar motivations to FLAIR — identifying perturbation effects and updating policy. I encourage the authors to explicitly position FLAIR with respect to this body of work, e.g., explain whether FLAIR offers better scalability or any other improvements

5. In page 45, It mentioned that "These values were determined by maximising the log-marginal-likelihood [80] after manually driving the robot for a given amount of time. After this step, we freeze the parameters throughout all the experiments to improve the speed of the model." My concern is: Isn't FLAIR supposed to be fully online? This sentence implies some offline hyperparameter tuning that contradicts earlier claims. I suggest revising to clarify.

(Remarks on code availability)

I have no concerns about the code.

Reviewer #2

(Remarks to the Author)

Summary of the work: The paper introduces FLAIR, a fast, online residual-learning layer that augments an existing low-level controller to restore robot operability, i.e., command-following when robots face unmodeled perturbations. FLAIR learns a command-to-behavior correction from onboard data only and updates its model. It combines a global adaptation with a local adaptation using Gaussian Processes to model state/command-dependent effects. Experiments on a tracked mobile robot across multiple circuits and perturbation settings show a large reduction in tracking error and completion time versus a driver-only baseline. The paper also introduces introspection capabilities, i.e., an understanding of possibly present perturbations and their effect on the robot performance. However, these effects are described just by simple polynomial models.

Strengths:

1. The work targets fast recovery from unmodeled disturbances during deployment, a crucial gap between lab-tested controllers and messy field conditions.
2. The proposed algorithm wraps around an existing controller. It does not require redesigning the inner loop or extra sensors/external compute, and is generally amenable to any controller.
3. From the result it is clear that the end-to-end model refresh time is on the order of hundreds of milliseconds, making the application suitable for many mobile platforms.
4. There is a sensible split between global and local effects and leverages structure of differential-drive kinematics.
5. The work has a very strong empirical scope and also compares against a breadth of baselines.

Weaknesses:

Despite the many strengths, there are some very important weaknesses, most of which stem from the presentation of the work and the problem formulation.

1. Currently, as it is written, there is significant ambiguity in what exactly is being learned. The authors write that "FLAIR learns a hierarchical command-behaviours mapping that characterises the effect of perturbations on the system kinematics and the behavior of the robot." This is very vague. To make it very clear, I would suggest specifying the explicit learned function(s). It would be even beneficial if the authors could include a problem setup subsection or a section.
2. The core idea is very reasonable, but the paper does not isolate what is technically new versus known (e.g., parametric global bias + nonparametric residual; bounded buffer; change detection + refit etc.). As written, the contribution reads as a collection of sensible engineering choices rather than a unifying theorem or complexity/optimality result.
3. The authors mention that the robot is operable when the requested velocities correspond to the observed velocities. From what I understand, that assumes that (i) the operator's "desired behavior" is expressed solely through velocity commands, and the vehicle exhibits no nominal dynamics or latency. These assumptions obviously collapse high-level command-following to simple algebraic equality and importantly, sidestep transient response, stability margins, and variants of safety constraints.
4. The class of disturbances that the algorithm deals with is a little unclear. Specifically, in the "Methods" sections authors explicitly model perturbations as scaling of wheel/track velocities. This class of perturbations is also described in "Detailed Perturbations", which also mentions an interesting extension to state-dependent scaling. But wind, which is extensively used in experimentation, is unlikely to be solely state-dependent over a mission of nontrivial length (even if we accept it to result in scaling of wheel/track velocities, which is a little counterintuitive as well).
5. Is the minimization in Eq. 2 missing a weighting matrix? The identity weighting between linear (m/s) and angular (rad/s) velocity seems ill-posed/scale-dependent.
6. Adding on to Comment #5, the computation of Eq. 2 online, unless the authors use some particular structure-aware tricks, requires either access to (i) a continuous optimizer with gradients, for a differentiable f , and/or (ii) a structured search over a discretized command set. The text does not state which method is used, what the computational budget is, or how the work would avoid any nonconvex local minima. This issue is central to real-time viability.
7. Nothing in argmin of Eq. 2 prevents the solution from violating operator speed limits to reduce error. It would make sense to add explicit constraints.
8. The no pre-training claim is contradicted by hyperparameter learning on 1000 pre-collected points. The authors tune GP length-scales/state parameters by maximizing marginal likelihood on an offline dataset. By definition, that IS pre-tuning.

Unless I misunderstood something, the authors' claim should be that the algorithm requires no "task-specific pre-training". This claim about pre-training appears multiple times, so it seems very important to address it.

9. GP derivation is written with $m(a)$, with no dependence on s . However, later the authors say the kernel uses (a,s) as inputs. This is inconsistent notation; please fix and state the actual regressor $z=(a,s,\text{maybe history})$.

10. The work also claim LS on Eq. (7) (GPU) and GP training from scratch. It is worth clarifying if the polynomial coefficients are updated by LS outside the GP, and then used as a fixed mean for the GP update, or if they are jointly re-estimated inside a marginal-likelihood optimization.

11. Earlier sections emphasized ~225 ms (line 34) end-to-end adaptation; here the authors claim "<50 ms total learning time" (line 803) for GP training with 1000 points. I could not find any specifications of hardware, optimizer iterations and a discussion as to whether change-detection resets trigger hyperparameter re-optimization (expensive). In either case, the numbers seem conflict.

12. State variable s is under-specified and mostly treated as 1-D. Who chooses s ? Is it learned, or manually selected in each scenario? Isn't a single scalar s too restrictive for real robots, which are usually taken to live in some Euclidean space or manifold

13. Would it be possible to justify the choice of RBF and state the priors/constraints on length-scales and noise? From what I know, GP can over-smooth and underfit local perturbations (ironically what local adaptation is meant to fix). Given this, wouldn't spectral mixtures be more appropriate?

14. (Line 846) The agreement metric lacks normalization and units: I uses the predicted behavior for the adapted command a' ; ϵ uses the observed behavior for the unadapted command a . Subtracting them is not a model-residual in any classical sense. In the same paragraph, what is 0.33 measured against (units missing)?

15. If I am reading correctly, upon any change, the work empties the dataset. Wouldn't that lose valuable information about global structure that may still be valid? Wouldn't a more principled approach retain a global model and reset only local residuals? As is, it feels like the work risks oscillations between learn–reset–relearn.

16. Similarly, won't picking the lowest error among synchronized samples (even after trimming) bias the data toward under-estimating the disturbance; i.e., won't it delay detection and yield over-optimistic residuals?

17. I am still unclear on the theoretical grounding, or even a somewhat rigorous intuition, behind the claimed fast adaptation of FLAIR. Namely, the grid of FIFO micro-buffers definitely improves coverage but it also naturally ignores the temporal non-stationarity the work is trying to track. There does not seem to be any age weighting or forgetting factor within a cell, so stale points from rarely used commands can easily dominate after a change. How is the method able to adapt as fast?

(Remarks on code availability)

The codebase is well structured and generally straightforward to navigate, with a clean separation between the simulation pipeline and the real-robot implementation. However, much of the repository--everything outside the Hexapod simulation--relies on the specialized hardware described in the paper. The authors themselves point out that the ROS topic names must be adapted to match a given setup, which means that reproducing the real-robot results is only feasible for labs with comparable sensors, computing hardware, and motion-capture systems.

Because of this constraint, I was unable to validate the hardware experiments and instead focused my review on the simulation results. The Hexapod example executed without major issues, though performance improvements were modest on an Apple M-series CPU. When run on an NVIDIA A100, however, the results aligned well with the performance reported in the paper. The most significant difficulties arose during environment setup. Although the project's README specifies Python 3.10, I encountered repeated incompatibilities with JAX at that version. I ultimately had success by upgrading to Python 3.11, which likely resolved a mismatch between the Brax and JAX versions. This is consistent with the fact that recent JAX releases have set Python 3.11 as the minimum supported version (see <https://github.com/google/jax/releases>). Once this adjustment was made, the simulation pipeline proved to be both a useful and reproducible resource. By contrast, while the real-robot pipeline appears well engineered, its utility for the broader community is naturally constrained by the specialized hardware requirements. Introducing tighter version pinning for JAX, Brax, and QDax--or providing a lockfile (conda/uv/poetry)--would go a long way toward smoothing setup and improving the overall accessibility of the repository.

Reviewer #3

(Remarks to the Author)

(Remarks on code availability)

The codebase is well structured and generally straightforward to navigate, with a clean separation between the simulation pipeline and the real-robot implementation. However, much of the repository--everything outside the Hexapod simulation--relies on the specialized hardware described in the paper. The authors themselves point out that topic names must be

adapted to match a given setup, which means that reproducing the real-robot results is only feasible for labs with comparable sensors, computing hardware, and motion-capture systems.

Because of this constraint, I was unable to validate the hardware experiments and instead focused my review on the simulation results. The Hexapod example executed without major issues, though performance improvements were modest on an Apple M-series CPU. When run on an NVIDIA A100, however, the results aligned well with the performance reported in the paper. The most significant difficulties arose during environment setup. Although the README specifies Python 3.10, I encountered repeated incompatibilities with JAX at that version. I ultimately had success by upgrading to Python 3.11, which likely resolved a mismatch between the Brax and JAX versions. This is consistent with the fact that recent JAX releases have set Python 3.11 as the minimum supported version (see <https://github.com/jax-ml/jax/releases>). Once this adjustment was made, the simulation pipeline proved to be both a useful and reproducible resource. By contrast, while the real-robot pipeline appears well engineered, its utility for the broader community is naturally constrained by the specialized hardware requirements. Introducing tighter version pinning for JAX, Brax, and QDax--or providing a lockfile (conda/uv/poetry)--would go a long way toward smoothing setup and improving the overall accessibility of the repository.

Version 1:

Reviewer comments:

Reviewer #1

(Remarks to the Author)

This work introduces FLAIR, an online hierarchical adaptation method that adjusts robot commands to recover operability in the presence of perturbations. The method combines global linear adaptation and local residual modeling using Gaussian Processes. The results are promising, showing significant recovery across various perturbation types.

The authors have addressed most of my concerns in a careful and constructive manner, and I appreciate the detailed clarifications provided in both the revised manuscript and the rebuttal.

Regarding Concern 1, my original intention was not to question the effectiveness of the proposed robustness mechanisms within the evaluated benchmarks. Rather, the concern was to highlight an inherent limitation of the approach with respect to robustness under more general real-world conditions. While the robustness track and the Soft Outlier Rejection mechanism demonstrate that the method performs well for the tested classes of perturbations, they do not establish robustness guarantees for broader categories of disturbances that are highly nonstationary, subtle, or discontinuous. The authors' explanation clarifies that the online learning mechanism remains effective as long as the encountered perturbations remain within the modeling assumptions. Under this interpretation, I find the clarification acceptable, provided that the scope of robustness claims is appropriately bounded.

For Concern 2, the authors explain in the rebuttal that a perturbation cannot, by definition, be simultaneously too small to trigger the reset mechanism and large enough to severely impair operability. While this assumption may hold in specific experimental or task-dependent settings, such situations can arise in practical deployments, where moderate but persistent mismatches accumulate over time without triggering explicit resets. That said, the discussion of replay buffer dynamics and eventual adaptation partially addresses this concern, even though transient safety implications in such scenarios remain an open question.

Overall, while the method necessarily exhibits limitations and its robustness should not be overgeneralized beyond the evaluated scenarios, I acknowledge that addressing all possible real-world disturbances is beyond the scope of a single study. Within its stated assumptions and experimental scope, the work presents a meaningful contribution by demonstrating a practical hierarchical online adaptation framework and by clearly articulating its mechanisms and empirical behavior.

(Remarks on code availability)

Reviewer #2

(Remarks to the Author)

I appreciate the effort in addressing each of the comments and in updating the paper accordingly (especially the new problem statement, clearer perturbation descriptions, and implementation details). Overall, I am satisfied with the clarifications, and I think the revised version will read much more clearly and reflect the intended scope of FLAIR.

A few small follow-up questions/suggestions that might be worth addressing briefly:

1. Since the current implementation assumes matched, actuator-channel perturbations via track scaling (even for wind), I would suggest adding a short sentence in the Limitations explicitly stating that true unmatched disturbances (e.g., lateral forces independent of actuation) are out of scope for this version of FLAIR?
2. For the 101x101 command grid, have the authors observed any sensitivity to this resolution in practice, and could the authors add a brief remark on why this particular grid size was chosen (e.g., trade-off between coverage and compute)?
3. It could help readers if the authors could explicitly say that the optimistic choice of the lowest-error synchronized sample is an intentional safety-oriented heuristic (trading bias for conservative adaptation), so it's clear this is by design rather than a bias.

(Remarks on code availability)
No further comments.

Reviewer #3

(Remarks to the Author)

(Remarks on code availability)

The updated code works well and the authors did a good job addressing all the comments.

2. Answers to Reviewer 1

We now address the comments raised by Reviewer 1 and highlight the corresponding changes in the paper.

RC: *1. [...] Such behaviors may violate the smoothness assumptions in both Taylor approximations and GPs, and potentially cause misleading generalizations. I am concerned that in these cases, online modeling could deteriorate performance, especially under rapid perturbation shifts that are too subtle for the model reset condition to trigger.*

AR: We agree with the reviewer that rapid perturbations can deteriorate the performance of online model-based methods when no filtering is applied. This concern directly motivates the Robustness track in our experiments, which includes very short perturbations such as sudden slippage and strong vibrations, and aims to study the robustness of FLAIR to such perturbations. This robustness comes from our Soft Outlier Rejection module, which filters out outlier points, such as the ones induced by rapid perturbations, before they are used for learning the model. Our results on the Robustness track illustrate the performance of this approach. Following the reviewer comment, we have revised the manuscript to clarify this point, describe the properties of the Robustness track, and discuss FLAIR's ability to handle these behaviours (attached below). We also emphasize that combining FLAIR with approaches designed to learn non-stationary functions is a promising direction for future work.

Although Gaussian Processes are data-efficient, they rely on smoothness assumptions of the learned function, which can be violated when encountering noisy data points, caused by collisions, sensory failures, or sudden ground failures. By leveraging a Soft Outlier Rejection mechanism, we are able to filter out highly variable data points that could hinder the learning of the Gaussian Process. We demonstrate the effectiveness of this mechanism in the Robustness circuit, showing that FLAIR can filter out noisy datapoints and keep a strong level of robustness. We note that FLAIR does not actively learn non-stationary functions but rather relies on resetting mechanisms and filtering of training datapoints. We leave the exploration of including models that can learn non-stationary functions to future work.

RC: *2. The method defines a model agreement threshold >0.33 to reset and relearn a new model. However, if the real-world perturbation causes moderate but persistent disagreement, just below the threshold, the system may continue using an outdated model, resulting in compounding control errors. There is no theoretical or empirical analysis of how sensitive the system is to sub-threshold dynamics, nor whether this can lead to performance being worse than non-adaptive baselines in some conditions. This safety-critical edge case deserves attention.*

AR: While we agree with the reviewer that perturbations that would not trigger a strict reset exist, we disagree that they constitute a safety-critical edge for two reasons. First, by definition, a perturbation cannot be at the same time small enough not to trigger the reset mechanism and large enough to drastically impact the operability of the robot. Second, even if these perturbations do not trigger an explicit reset through the Continuous Reset Identification module, they will still eventually become predominant in the replay buffer through the double FIFO mechanism and the Soft Outlier Rejection mechanism. Thus, such perturbations will still be addressed by the model update as if it had reset, although the compensation might take more time than with a standard reset. In the worst possible case, where the perturbation affects the entire command space and none of the datapoints trigger the Soft Outlier Rejection, 500 datapoints would need to be replaced for the new perturbation to become the majority, which would require 50 seconds. However, it is very unlikely that a perturbation satisfying these conditions would fail to trigger an explicit reset, and a more realistic scenario is that it would affect only a limited number of neighbourhoods. Each neighbourhood would require 1.5 seconds to be fully updated, again assuming that the Soft Outlier Rejection does not reduce this time, which is

itself unlikely. In summary, in the worst-case scenario where a perturbation does not trigger a reset, it would still eventually reach the same model as if a reset had been triggered. We thank the reviewer for raising this interesting point. We added this clarification to the description of the reset mechanism (attached below), as we believe this will likely be interesting for other readers.

Note that in the case of a perturbation with only a mild impact that fails to reach this threshold, the dataset management mechanisms in FLAIR, described in the next sections, would still ensure that this new perturbation eventually becomes predominant in the replay buffer, thereby allowing the model update to handle it as if it had undergone a reset.

RC: *3. Although the results demonstrate solid performance, it's not obvious that FLAIR's performance is decisively better than simpler baselines in all cases. In several cases, L1 or LQR recover a substantial fraction of functionality (often >50%) without any learned model. These methods are intrinsically robust due to the control structure, while FLAIR depends on the correctness of learned models and online resets.*

AR: We agree with the reviewer's comment that FLAIR is dependent on the correctness of the learned model and its resets. However, we believe this is precisely one of the key contributions of this paper to show that a learning based method can demonstrate the same level of stability as methods like LQR and L1 while achieving higher performance. Our experiments have been designed to demonstrate a variety of scenarios to directly evaluate the performance and stability of the learned models and resets. To highlight further this aspect in the paper, we've added the snippet of explanations below at the beginning of the results section:

FLAIR has been designed to recover operability across a variety of tasks by learning the correct model online to compensate for perturbations. FLAIR reduces the error induced by perturbations by half when compared to our L1 and LQR baselines, demonstrating that it is able to learn the correct model online and correctly reset it. In comparison, L1 or LQR can, in some cases, reduce the error by $\approx 50\%$, however we argue that this is insufficient in scenarios requiring high manoeuvrability and that any reduction in perturbation is beneficial to the operator. These results show that a learning based method is able to outperform methods such as L1 and LQR by halving their tracking error.

RC: *4. In the RL literature, several works (e.g., Residual RL, Adaptive RL with online fine-tuning, Meta-RL) handle nonstationarity by combining offline policies with online correction or adaptation layers. These share similar motivations to FLAIR — identifying perturbation effects and updating policy. I encourage the authors to explicitly position FLAIR with respect to this body of work, e.g., explain whether FLAIR offers better scalability or any other improvements*

AR: We agree with the reviewer that these bodies of work are highly relevant to the focus of FLAIR, and that the paper currently does not sufficiently position itself with respect to them. We wish to clarify that while they solve similar problems (i.e. online adaptation using learning), these methods stay distinct. Indeed, they aim to find ways to adapt a specific policy, first trained offline to control a specific robot. FLAIR, on the contrary, aims to be a high-level module that performs fully online and is agnostic to the underlying low-level controller. We added these bodies of work to our Related Work Section (see below).

Some research directions in RL are closely related to our work, such as Continual RL [9, 1], Meta-RL [6, 2, 5], and Residual RL [13, 8]. However, these approaches differ from the motivation behind FLAIR in several important aspects. First, these methods typically aim to learn a full controller that can be directly applied to the robot, without relying on an existing low-level controller or on human inputs. FLAIR adopts a different perspective by introducing a hierarchical layer applied on top of an existing

low-level controller rather than replacing it, and adapting incoming human command. Second, most of these approaches require several seconds to minutes of data on the new task to perform adaptation. A key property of FLAIR is that it adapts in less than a second, from only a few datapoints.

To illustrate the limitations of such data-driven fine-tuning methods in real online scenarios, we include a Residual RL baseline in our experiments (described in detail in Section??). This baseline uses the same data processing pipeline and hardware as FLAIR, but learns a residual controller online using RL on the robot. In our experiments, this led to unsafe behaviors such as sudden accelerations or sudden backward movements, requiring the trials to be stopped. This result illustrates the challenges of applying online residual RL in the presence of perturbations and clarifies the conditions under which FLAIR provides reliable adaptation.

More generally, FLAIR is designed to complement any low-level controller, including RL-trained controllers. Its hierarchical structure allows it to be integrated into RL-based pipelines, where it can be applied at inference time as an online adaptation mechanism that adjusts operator inputs. Recent work has also highlighted the difficulties faced by gradient-based learning approaches, including RL, in continual or lifelong learning settings [4]. FLAIR offers an alternative direction for achieving rapid adaptation to perturbations without relying on continual gradient updates.

RC: 5. In page 45, It mentioned that “These values were determined by maximising the log-marginal-likelihood [80] after manually driving the robot for a given amount of time. After this step, we freeze the parameters throughout all the experiments to improve the speed of the model.” My concern is: Isn’t FLAIR supposed to be fully online? This sentence implies some offline hyperparameter tuning that contradicts earlier claims. I suggest revising to clarify.

AR: We thank the reviewer for raising this point. We would like to clarify that the parameters optimized through this procedure are robot-wide constants that only need to be optimized once and are not specific to an ongoing impairment nor contain any information about an "upcoming" impairment. Our assumption is that the operator can drive the robot for a few seconds under nominal, non-perturbed conditions (e.g. before the deployment). This short period allows the collection of a small dataset used to optimize these parameters, which are then fixed for the remainder of the operation and for all the successive impairment scenarios. We have clarified this point in the revised text as shown below.

These values are determined by maximising the log-marginal-likelihood [80] using data collected while the robot operates under non-perturbed conditions. To collect these data points, we assume the robot is initially operated under non-perturbed conditions for a few seconds using the pipeline described in this paper. After these few seconds of initial operation, we freeze the parameters throughout all the experiments, to improve the speed of the model.

3. Answers to Reviewer 2 and 3

We now address the comments raised in the co-review of Reviewers 2 and 3. We also highlight the corresponding changes in the paper in our answers.

RC: *The paper also introduces introspection capabilities, i.e., an understanding of possibly present perturbations and their effect on the robot performance. However, these effects are described just by simple polynomial models.*

AR: We agree with the reviewer, however, our approach can easily be generalised to any other regression algorithms: single linear, piece-wise linear, or Neural Networks. However, we deliberately choose to focus on polynomial models because they allow better interpretability and explainability for the operator, while staying powerful estimators. We added this explanation to the main text in Section 3.5. Introspection, also below.

Although various regression techniques (i.e. Kernel regression, Neural Networks, Gaussian Processes) could be used to learn this function, we prioritize interpretability and transparency, and therefore employ a polynomial model, that balances expressive power with ease of visualization and explanation for the operator.

RC: *1. [...] To make it very clear, I would suggest specifying the explicit learned function(s). It would be even beneficial if the authors could include a problem setup subsection or a section.*

AR: We would like to thank the reviewers for this excellent suggestion. We added a small problem setup section to the introduction to clarify that we are learning a function f that represents the impact of the perturbation on the behaviour of the robot given a command a and a state s .

Problem Statement Perturbations acting upon a robot alter its behaviour b (in our case the velocities v_x and ω_z) for a given command a . When faced with such perturbations, operators attempt to learn a mental model on how to change actions a into a' to obtain the desired behaviour b to regain operability (i.e. obtaining desired behaviour b through a command). In this work, we formalize a perturbation as a function $f(a, s) = b'$ that modifies the robot's behaviour when actions a are executed in state s under the perturbation. Our goal is to learn $f(a, s)$ in order to counteract the perturbation and augment the operator:

$$f: A \times S \rightarrow B' \tag{1}$$

RC: *2. The core idea is very reasonable, but the paper does not isolate what is technically new versus known (e.g., parametric global bias + nonparametric residual; bounded buffer; change detection + refit etc.). As written, the contribution reads as a collection of sensible engineering choices rather than a unifying theorem or complexity/optimality result.*

AR: We thank the reviewer for this insightful comment, and we agree that we haven't well summarized the contributions (in contrast to sensible engineering choices). We have added a paragraph to the introduction to highlight that our core contribution is the insight that leveraging a hierarchical layer helps to learn perturbation models in an online fashion. The manuscript itself explains how this hierarchical adaptation has been executed, with all the technical details, which include all the technical elements that have been highlighted in the comment.

This paper demonstrates that the hierarchical addition of an abstract layer with spatio-temporal data processing on top of the robot’s original black-box controller enables rapid adaptation of differential-drive robots to a variety of perturbations. The experiments consider both deployed tracked vehicles and simulated hexapod robots, facing perturbations including low-friction grounds, scaled motor torques, and external perturbations such as an artificial wind (illustrated in Figure 2). Our results show not only that a learning-based controller can adapt to external perturbation in an online manner using exclusively onboard compute, recovering 74.7% of the operability of the unperturbed robot, but also outperforms both optimal control and adaptive control baselines in our experiments by a factor of 2. Furthermore, by taking a hierarchical approach to adaptive control, we enable introspection capabilities, using a simple (e.g., polynomial) top-level model to represent the effect of perturbations.

RC: *3. The authors mention that the robot is operable when the requested velocities correspond to the observed velocities. From what I understand, that assumes that (i) the operator’s “desired behavior” is expressed solely through velocity commands, and the vehicle exhibits no nominal dynamics or latency. These assumptions obviously collapse high-level command-following to simple algebraic equality and importantly, sidestep transient response, stability margins, and variants of safety constraints.*

AR: We would first like to clarify that expressing the operator’s intended behavior through velocity commands is standard practice in robotics, rather than a modeling assumption introduced by FLAIR. For instance, the GVR-Bot used in our experiments comes from the manufacturer with a velocity-controlled joystick interface. Similarly, commercially available quadrupeds (e.g., Unitree, Boston Dynamics) are typically controlled using velocity-based inputs. Thus, this design choice reflects common practice in the field rather than any assumption specific to FLAIR.

To address the reviewer’s concern regarding safety, we wish to emphasize that FLAIR functions as a hierarchical layer designed to operate on top of a low-level controller that is specific to the robot at hand. The responsibility for handling the transient response, maintaining stability margins, and enforcing any other constraints required for safe control of the robot remains entirely with this low-level controller, which has been explicitly designed for the particular platform. FLAIR, in contrast, operates at the level of the operator’s inputs, transparently adjusting them so that the user’s original intentions are faithfully executed. Accordingly, as noted by the reviewer, FLAIR primarily focuses on managing nominal system dynamics and latency, while leaving core safety and stability guarantees to the underlying controller. In our work, we stress that it is crucial to match commands with corresponding behaviours to overcome these issues. We have highlighted our synchronisation method in the Supplementaries (Additional Details on Dataset Creation - Synchronisation in Data Point Creation).

RC: *4. The class of disturbances that the algorithm deals with is a little unclear. Specifically, in the “Methods” sections authors explicitly model perturbations as scaling of wheel/track velocities. This class of perturbations is also described in “Detailed Perturbations”, which also mentions an interesting extension to state-dependent scaling. But wind, which is extensively used in experimentation, is unlikely to be solely state-dependent over a mission of non-trivial length (even if we accept it to result in scaling of wheel/track velocities, which is a little counter-intuitive as well).*

AR: We thank the reviewer for highlighting that the state-dependent perturbation is not well defined. We have improved our description of the perturbations based on this review and are confident this clarifies the points made by the reviewer.

We added the following details to the methods section, highlighting that the wind is modelled as a continuous wind coming from a fixed direction that can exert forces on the robot via a sail, making it rotation-dependent.

We agree that, in practice, wind direction is unlikely to remain constant throughout deployment, which is precisely why FLAIR includes a continuous scenario identification mechanism that can reset the modelling when changes occur. To highlight the benefit of each individual mechanism of FLAIR, we’ve decided to split the perturbation classes into: (i) a static perturbation scenario (used to evaluate baseline adaptation), (ii) a dynamic perturbation which changes over the course of the deployment (used to test the continuous scenario identification), and (iii) a perturbation that changes over time in a state-dependent manner (also testing the continuous scenario identification but also dependence on a state variable that changes based on the robot’s observations) and (iv) a robustness track (testing the Soft Outlier Rejection Mechanism and FLAIR’s ability to learn and reset in highly noisy environments).

Finally, the third type of perturbation changes over time but is also state-dependent. Examples include continuous wind blowing into the robot (rotation-dependent) or closeness to borders for Bernoulli forces (position-dependent).

We have added the following to the Detailed Perturbations Section:

The assumption regarding the wind is that it is continuously blowing into a fixed direction. This makes the perturbation state-dependent for the robot since the sail of the robot would have a different area of attack for the wind, directly influencing the force that is applied to the robot. In our closed room, the wind is achieved by varying the scaling factors $d_{left}(s)$ and $d_{right}(s)$, which are dependent on the state $s = \text{yaw}$ of the robot’s orientation. This means that if the wind is stronger (i.e. perpendicular to the robot’s sail) then the scaling is the highest. We decrease the scaling linearly with the rotation and disable the scaling if the robot is parallel to the wind. Importantly, since the perturbation is virtual, FLAIR doesn’t have access to the underlying function but needs to learn the perturbations via the data it collects.

RC: *5. Is the minimization in Eq. 2 missing a weighting matrix? The identity weighting between linear (m/s) and angular (rad/s) velocity seems ill-posed/scale-dependent.*

AR: We appreciate the reviewer’s comment. We updated Eq.2 to include a weighting matrix. In our case, this matrix is an identity matrix because the overall scale of desired behaviours matches the range of the manufacturer out of the box.

$$\begin{aligned} \pi(s, b) = \arg \min_{a'} & \quad \cdot \cdot D(f(a', s) - b) \cdot \cdot \\ \text{subject to} & \quad -2 < a' < 2 \end{aligned} \quad (2)$$

D is a diagonal matrix with per-dimension scaling factors. In our case, we use the Identity matrix.

RC: *6. Adding on to Comment 5, the computation of Eq. 2 online, unless the authors use some particular structure-aware tricks, requires either access to (i) a continuous optimizer with gradients, for a differentiable f, and/or (ii) a structured search over a discretized command set. The text does not state which method is used, what the computational budget is, or how the work would avoid any nonconvex local minima. This issue is central to real-time viability.*

AR: We thank the reviewer for this helpful comment. This information was indeed missing from the manuscript. Our proposed FLAIR method discretizes the command space to compute Equation 2, and we have now clarified this point in the revised text as follows.

In the unperturbed case, the model f is an identity function of the command and behaviour, as the executed command directly yields the desired behaviour. To speed up the optimisation in Equation 3, we uniformly discretize the command space by a grid of size 101×101 ranging from -2 to 2 for both linear and angular velocity. The function f is then evaluated over all grid points to identify the optimal action. This brute-force optimisation of 10, 201 commands is done online and in parallel on the GPU due to the optimised software.

RC: *7. Nothing in argmin of Eq. 2 prevents the solution from violating operator speed limits to reduce error. It would make sense to add explicit constraints.*

AR: We agree with the reviewer and added an explicit constraint in Equation 2. This change is shown in the new version of Equation 2 in answer to point 5. The constraint of -2 and 2 is put in place by the manufacturer, which is why it was implicitly enforced in the code. We agree that this was missing from the paper, and we appreciate that it was pointed out by the reviewer.

RC: *8. The no pre-training claim is contradicted by hyper-parameter learning on 1000 pre-collected points. The authors tune GP length-scales/state parameters by maximizing marginal likelihood on an offline dataset. By definition, that IS pre-tuning. Unless I misunderstood something, the authors' claim should be that the algorithm requires no "task-specific pre-training". This claim about pre-training appears multiple times, so it seems very important to address it.*

AR: We agree with the reviewer that this was unclear. We would like to clarify that the parameters optimized through this procedure are robot-wide constants that only need to be optimized once and are not specific to an ongoing impairment nor contain any information about an "upcoming" impairment. Our assumption is that the operator can drive the robot for a few seconds under nominal, non-perturbed conditions (e.g. before the deployment). This short period allows the collection of a small dataset used to optimize these parameters, which are then fixed for the remainder of the operation and for all the successive impairment scenarios. We have clarified this point in the revised text as shown below. Additionally, following the reviewer's suggestion, we have also replaced "pre-training" with "task-specific pre-training" in the text.

These values are determined by maximising the log-marginal-likelihood [10] using data collected while the robot operates under non-perturbed conditions. To collect these data, we assume the robot is initially operated under non-perturbed conditions for a few seconds using the pipeline described in this paper. After these few seconds of initial operation, we freeze the parameters throughout all the experiments, to improve the speed of the model.

RC: *9. GP derivation is written with $m(a)$, with no dependence on s . However, later the authors say the kernel uses (a,s) as inputs. This is inconsistent notation; please fix and state the actual regressor $z=(a,s,\text{maybe history})$.*

AR: We agree with the reviewer and updated our equations accordingly as follows.

$$f(s', a') \sim GP(m(s', a'), k((s, a), (s', a')))$$

with the predictive mean:

$$\hat{f} = m(s', a) + K((s', a), (s, a))K_{\gamma}^{-1} (b - m(s, a))$$

RC: 10. The work also claim LS on Eq. (7) (GPU) and GP training from scratch. It is worth clarifying if the polynomial coefficients are updated by LS outside the GP, and then used as a fixed mean for the GP update, or if they are jointly re-estimated inside a marginal-likelihood optimization.

AR: We agree with the reviewer and clarified this point in the text as follows.

To combine both the local and global adaptation to perturbations, FLAIR uses the Equation 8 as mean function $m(s, a)$ to our Gaussian process. By optimising Equation 8, FLAIR can update its beliefs around the global impact of the perturbations while adjusting for local disparities with the Gaussian process model. The polynomial parameters in Equation 8 are only solved via the least squares method, and then used as fixed GP mean.

RC: 11. Earlier sections emphasized 225 ms (line 34) end-to-end adaptation; here the authors claim "<50 ms total learning time" (line 803) for GP training with 1000 points. I could not find any specifications of hardware, optimizer iterations and a discussion as to whether change-detection resets trigger hyperparameter re-optimization (expensive). In either case, the numbers seem conflict.

AR: We agree with the reviewer on this point and have clarified the text accordingly. Regarding the hardware setup, we provide a detailed description of the full system in the Supplementary Materials:

To run our onboard computation, we are using an NVIDIA Jetson AGX Orin 32GB that we have mounted on top of the robot.

Regarding the reported times, we acknowledge that the distinction between the different values was not sufficiently clear. The value of < 50ms corresponds to the learning time of the GP (only the training of the GP on the GPU without any of the data filtering, updates of commands or least-squares adaptation). We have added a detailed breakdown of the times in the Supplementary Materials:

To leverage GPU acceleration, the model training of FLAIR is implemented using the Jax [3] library, which makes use of XLA compilation [12]. Our implementation of Gaussian process is based on the structure of the GPJax [11] library, redesigned and optimised for our usage. Our implementation reduces the total model training and deployment time to less than 225ms (composed of ≈ 67 ms to 215ms of Gaussian Process Training and Least Squares Optimisation together + ≈ 9 ms updating the controller commands + ≈ 3 ms for data filtering). This is less than the median human reaction time for visual stimuli [7].

Additionally, we have clarified where the 225 ms come from in the main paper and have removed the 50ms claim in order to make it more comprehensive.

We parallelised and optimised our code to reduce the total learning and execution time on our on-device GPU (see Hardware Details in Supplementary Materials) to less than 225 ms which includes the learning of the GP, data filtering, command updates and the least-squares optimisation for the polynomial hyper-parameters in d' (more details on the implementation can be found in the Supplementary Materials 4). The hyper-parameter optimisation only happens once during the first few seconds of operation, and is

not re-run at any point during the deployment of FLAIR. Even after a reset, the same hyper-parameters are used and the only learning that happens is the local (GP) and global (Least Squares) optimisation.

RC: *12. State variable s is under-specified and mostly treated as 1-D. Who chooses s ? Is it learned, or manually selected in each scenario? Isn't a single scalar s too restrictive for real robots, which are usually taken to live in some Euclidean space or manifold*

AR: We agree with the reviewer that a 1-D state variable might not contain all the information required to solve different tasks, but in our case, we selected the z-axis rotation as the state variable manually, and didn't change it across all our experiments. Future directions could include a more general approach to include higher-dimensional state-spaces, which might be more flexible across different perturbations or an automatic identification outside of FLAIR, which dimension of the state might be of relevance. We added a section to the Global Adaptation Section:

Since perturbations are not constant but may vary with the state, d' can be a function of the relevant state dimension s . In our case, we assume that an engineer has selected the relevant state information s , such as the displacement or the rotation of the robot, depending on the perturbations. Here, we use the rotation across all our experiments (except the treadmill). We approximate the global command-behaviour mapping $d'(s)$ with a Taylor expansion up to the third degree with parameters $\alpha, \beta, \gamma, \delta$. We decided to use a third-degree polynomial since the Bernoulli force used during the LINC experiments is quadratic, and to cover any complex non-linear perturbations (e.g. complex drag), we add a third degree to the polynomial that can be used to approximate perturbations. The function could be approximated with other methods (e.g. Deep Learning) or another Taylor series. This could be useful to integrate higher-dimensional state information into the function $d'(s)$, but in our experiments, we did not explore the function choice further, given the positive results with the Taylor series.

RC: *13. Would it be possible to justify the choice of RBF and state the priors/constraints on length-scales and noise? From what I know, GP can over-smooth and underfit local perturbations (ironically what local adaptation is meant to fix). Given this, wouldn't spectral mixtures be more appropriate?*

AR: While we agree that our method could benefit from more complex modelling, we did not study kernel selection in detail, as the RBF kernel proved effective in our experiments. This is likely due to the Soft Outlier Rejection Mechanism, which helps remove highly noisy points, hence making the training with a smooth kernel easier. We have clarified this point in the paper and explicitly highlighted it as potential future work (see below).

Different hyper-parameters have been tried (e.g. learning them on different terrains) with little difference in adaptation performance. We did not experiment with alternative kernels as the RBF kernel performed very well in our experiments. The Soft Outlier Rejection mechanism likely removes a lot of the non-smooth data-points which probably helps the GP learn well with a RBF kernel. Other kernels might be better suited for specific use cases or robot types, but we leave this investigation for future work, as it falls outside the scope of the present study.

RC: *14. (Line 846) The agreement metric lacks normalization and units: l uses the predicted behavior for the adapted command a' ; ϵ uses the observed behavior for the unadapted command a . Subtracting them is not a model-residual in any classical sense. In the same paragraph, what is 0.33 measured against (units missing)?*

AR: We agree with the reviewer and added normalisation constants that are behaviour-dependent:

[]

Both error vectors are normalised by a characteristic behaviour scale $\sigma_b = 1.25$ in our case:

$$\tilde{\mathbf{t}} = \frac{|\mathbf{b} - \mathbf{b}'|}{\sigma_b}, \quad \tilde{\boldsymbol{\epsilon}} = \frac{|\mathbf{b}'_{\text{obs}} - \mathbf{b}|}{\sigma_b}.$$

Finally, FLAIR subtracts both errors to measure how much the reality (i.e. the system) deviates from what the model would have predicted, which we call model agreement ν :

$$\nu = \|\tilde{\mathbf{t}} - \tilde{\boldsymbol{\epsilon}}\|_2 \quad (3)$$

RC: *15. If I am reading correctly, upon any change, the work empties the dataset. Wouldn't that lose valuable information about global structure that may still be valid? Wouldn't a more principled approach retain a global model and reset only local residuals? As is, it feels like the work risks oscillations between learn–reset–relearn.*

AR: The reviewer's point is correct: whenever a change in perturbation is detected by the Continuous Scenario Identification module, it triggers a reset of the other modules, thereby emptying the dataset. While we understand the reviewer's concern, we wish to emphasize that this module is designed to trigger in face of significant perturbation changes, and thus all data points being removed correspond to an outdated perturbation. Additionally, in such a noisy setting, only a subset of the data points carries meaningful information, and there is no straightforward way to identify them. Extracting global structural information from data points from another perturbation would require complex learning modules capable of modeling this global context (e.g. periodicity). Finally, we wish to emphasize that, in this setting, data accumulate rapidly and FLAIR trains quickly, allowing the system to promptly compensate for the dataset reset.

RC: *16. Similarly, won't picking the lowest error among synchronized samples (even after trimming) bias the data toward under-estimating the disturbance; i.e., won't it delay detection and yield over-optimistic residuals?*

AR: We believe that, on the contrary, being conservative here is key to the proper operation of the system. We wish to clarify that the estimation of perturbations' effect runs multiple times per second, each time accumulating more data that improves its estimate. Therefore, it is safer to favor a more conservative and iterative improvement, rather than an aggressive but potentially unstable response.

RC: *17. I am still unclear on the theoretical grounding, or even a somewhat rigorous intuition, behind the claimed fast adaptation of FLAIR. Namely, the grid of FIFO micro-buffers definitely improves coverage but it also naturally ignores the temporal non-stationarity the work is trying to track. There does not seem to be any age weighting or forgetting factor within a cell, so stale points from rarely used commands can easily dominate after a change. How is the method able to adapt as fast?*

AR: We wish to clarify that we believe the proposed solution does account for temporal non-stationarity. First, there is a forgetting factor within the cells through the double FIFO mechanism, which removes both the oldest data points of each cell, and the oldest data points in the grid regardless of their cells. Second, in addition to the double FIFO mechanisms, our proposed approach also implements two further mechanisms: Soft Outlier Rejection and Continuous Reset Identification, both of which allow the model to account for temporal non-stationarity. The Soft Outlier Rejection is specifically responsible for detecting slow perturbation changes by "hiding" datapoints that do not fit the current prediction. This mechanism detects outliers coming from noises, collisions, or other unpredictable events, but it also allows to hide remaining datapoints from outdated

perturbations, thus making eventual transitions more efficient. Similarly, the Continuous Reset Identification aims to identify sudden changes in the perturbations, and to trigger a re-learning from scratch by immediately getting rid of the current dataset, thus effectively accounting for non-stationarity.

Finally, we would like to emphasise again that our use of JAX and an efficient GP implementation enables fast and frequent model updates (every 225 ms). This update rate allows the model to constantly account for varying conditions and non-stationarity.

RC: *I ultimately had success by upgrading to Python 3.11, which likely resolved a mismatch between the Brax and JAX versions. [...] Introducing tighter version pinning for JAX, Brax, and QDax—or providing a lockfile (conda/uv/poetry)—would go a long way toward smoothing setup and improving the overall accessibility of the repository.*

AR: We thank the reviewer for their thorough review of our code. We apologise for the version mismatch problem, and we have updated the repository to solve this conflict.

References

- [1] David Abel, André Barreto, Benjamin Van Roy, Doina Precup, Hado P van Hasselt, and Satinder Singh. A definition of continual reinforcement learning. *Advances in Neural Information Processing Systems*, 36:50377–50407, 2023.
- [2] Jacob Beck, Risto Vuorio, Evan Zheran Liu, Zheng Xiong, Luisa Zintgraf, Chelsea Finn, Shimon Whiteson, et al. A tutorial on meta-reinforcement learning. *Foundations and Trends® in Machine Learning*, 18(2-3):224–384, 2025.
- [3] James Bradbury, Roy Frostig, Peter Hawkins, Matthew James Johnson, Chris Leary, Dougal Maclaurin, George Necula, Adam Paszke, Jake VanderPlas, Skye Wanderman-Milne, and Qiao Zhang. JAX: composable transformations of Python+NumPy programs. 2018.
- [4] Shibhansh Dohare, J Fernando Hernandez-Garcia, Qingfeng Lan, Parash Rahman, A Rupam Mahmood, and Richard S Sutton. Loss of plasticity in deep continual learning. *Nature*, 632(8026):768–774, 2024.
- [5] Yan Duan, John Schulman, Xi Chen, Peter L Bartlett, Ilya Sutskever, and Pieter Abbeel. RL²: Fast reinforcement learning via slow reinforcement learning. *arXiv preprint arXiv:1611.02779*, 2016.
- [6] Chelsea Finn, Pieter Abbeel, and Sergey Levine. Model-agnostic meta-learning for fast adaptation of deep networks. In *International conference on machine learning*, pages 1126–1135. PMLR, 2017.
- [7] Aditya Jain, Ramta Bansal, Avnish Kumar, and KD Singh. A comparative study of visual and auditory reaction times on the basis of gender and physical activity levels of medical first year students. *International journal of applied and basic medical research*, 5(2):124–127, 2015.
- [8] Tobias Johannink, Shikhar Bahl, Ashvin Nair, Jianlan Luo, Avinash Kumar, Matthias Loskyll, Juan Aparicio Ojea, Eugen Solowjow, and Sergey Levine. Residual reinforcement learning for robot control. In *2019 international conference on robotics and automation (ICRA)*, pages 6023–6029. IEEE, 2019.
- [9] Khimya Khetarpal, Matthew Riemer, Irina Rish, and Doina Precup. Towards continual reinforcement learning: A review and perspectives. *Journal of Artificial Intelligence Research*, 75:1401–1476, 2022.
- [10] Radford M Neal. *Bayesian learning for neural networks*, volume 118. Springer Science & Business Media, 2012.
- [11] Thomas Pinder and Daniel Dodd. Gpjax: A gaussian process framework in jax. *Journal of Open Source Software*, 7(75):4455, 2022.
- [12] Amit Sabne. Xla : Compiling machine learning for peak performance, 2020.
- [13] Tom Silver, Kelsey Allen, Josh Tenenbaum, and Leslie Kaelbling. Residual policy learning. *arXiv preprint arXiv:1812.06298*, 2018.

1. Answers to Reviewer 1

RC: *The authors have addressed most of my concerns in a careful and constructive manner, and I appreciate the detailed clarifications provided in both the revised manuscript and the rebuttal. Regarding Concern 1, my original intention was not to question the effectiveness of the proposed robustness mechanisms within the evaluated benchmarks. Rather, the concern was to highlight an inherent limitation of the approach with respect to robustness under more general real-world conditions. While the robustness track and the Soft Outlier Rejection mechanism demonstrate that the method performs well for the tested classes of perturbations, they do not establish robustness guarantees for broader categories of disturbances that are highly nonstationary, subtle, or discontinuous. The authors' explanation clarifies that the online learning mechanism remains effective as long as the encountered perturbations remain within the modeling assumptions. Under this interpretation, I find the clarification acceptable, provided that the scope of robustness claims is appropriately bounded. For Concern 2, the authors explain in the rebuttal that a perturbation cannot, by definition, be simultaneously too small to trigger the reset mechanism and large enough to severely impair operability. While this assumption may hold in specific experimental or task-dependent settings, such situations can arise in practical deployments, where moderate but persistent mismatches accumulate over time without triggering explicit resets. That said, the discussion of replay buffer dynamics and eventual adaptation partially addresses this concern, even though transient safety implications in such scenarios remain an open question. Overall, while the method necessarily exhibits limitations and its robustness should not be overgeneralized beyond the evaluated scenarios, I acknowledge that addressing all possible real-world disturbances is beyond the scope of a single study. Within its stated assumptions and experimental scope, the work presents a meaningful contribution by demonstrating a practical hierarchical online adaptation framework and by clearly articulating its mechanisms and empirical behavior.*

AR: We appreciate the reviewer's recognition of our work as a meaningful contribution. We are pleased that our clarifications regarding the limitations have successfully addressed the reviewer's concerns and helped refine the scope of the contribution. We thank the reviewer for their feedback and the constructive dialogue throughout the review process.

2. Answers to Reviewer 2

RC: *Since the current implementation assumes matched, actuator-channel perturbations via track scaling (even for wind), I would suggest adding a short sentence in the Limitations explicitly stating that true unmatched disturbances (e.g., lateral forces independent of actuation) are out of scope for this version of FLAIR?*

AR: We agree with the reviewer that explicitly stating the scope of our experiments would provide helpful context for the reader. We tested track-scaled perturbations for the purpose of reproducibility (i.e. running the same experiment in sandy or windy conditions is not reproducible across two trials). Following this suggestion, we have added two sentences stating this point to the limitations in the Discussion section:

In this paper, we tested FLAIR on actuator-channel perturbations for reproducibility and accurate evaluation. While FLAIR was also evaluated on true (non-actuation-based) disturbances as part of the LINC program, true unmatched disturbances (e.g., lateral forces independent of actuation) are out of scope for this version of this paper.

RC: *For the 101x101 command grid, have the authors observed any sensitivity to this resolution in practice, and could the authors add a brief remark on why this particular grid size was chosen (e.g., trade-off between coverage and compute)?*

AR: We thank the reviewer for this observation. During our development, we evaluated alternative resolutions, including 51×51 points and 151×151 points. We found the 101×101 grid to be the most effective balance for our requirements. We have added the following sentence to the manuscript to clarify this choice.

101 points was chosen as a trade-off between the precision of 51 points and the computational cost of 151 points.

RC: *It could help readers if the authors could explicitly say that the optimistic choice of the lowest-error synchronized sample is an intentional safety-oriented heuristic (trading bias for conservative adaptation), so it's clear this is by design rather than a bias.*

AR: We agree with the reviewer that explicitly stating this design choice in the text clarifies the intent of the mechanism. We added the following sentence to the manuscript:

This is an intentional safety-oriented heuristic that guarantees conservative adaptation by favouring data points with lower command-behaviour error.